# Tritrophic metabolism of plant chemical defenses and its effects on herbivore and predator performance

Ruo Sun[1], Xingcong Jiang[2], Michael Reichelt[1], Jonathan Gershenzon[1], Sagar Subhash Pandit[3]*, Daniel Giddings Vassão[1]*

[1]Department of Biochemistry, Max Planck Institute for Chemical Ecology, Jena, Germany; [2]Department of Evolutionary Neuroethology, Max Planck Institute for Chemical Ecology, Jena, Germany; [3]Molecular and Chemical Ecology Lab, Indian Institute of Science Education and Research, Pune, India

**Abstract** Insect herbivores are frequently reported to metabolize plant defense compounds, but the physiological and ecological consequences are not fully understood. It has rarely been studied whether such metabolism is genuinely beneficial to the insect, and whether there are any effects on higher trophic levels. Here, we manipulated the detoxification of plant defenses in the herbivorous pest diamondback moth (*Plutella xylostella*) to evaluate changes in fitness, and additionally examined the effects on a predatory lacewing (*Chrysoperla carnea*). Silencing glucosinolate sulfatase genes resulted in the systemic accumulation of toxic isothiocyanates in *P. xylostella* larvae, impairing larval development and adult reproduction. The predatory lacewing *C. carnea*, however, efficiently degraded ingested isothiocyanates via a general conjugation pathway, with no negative effects on survival, reproduction, or even prey preference. These results illustrate how plant defenses and their detoxification strongly influence herbivore fitness but might only subtly affect a third trophic level.

**\*For correspondence:**
sagar@iiserpune.ac.in (SSP);
vassao@ice.mpg.de (DGV)

**Competing interests:** The authors declare that no competing interests exist.

## Introduction

The chemical defenses of plants constitute a major obstacle to herbivore feeding. Many plant defenses are glycosylated pro-toxins, such as glucosinolates, benzoxazinoids, cyanogenic glucosides, and iridoid glucosides, that are non-toxic by themselves, but liberate toxins after activation by glucosidases (*Morant et al., 2008*; *Pentzold et al., 2014*). Glucosinolates are activated specifically by β-thioglucosidases called myrosinases, which are stored separately from their substrates to avoid self-intoxication (*Bones and Rossiter, 1996*). Activation occurs upon plant damage, for example herbivore feeding, when compartments containing enzymes and the glucosylated pro-toxins are ruptured, leading to mixing of their contents and subsequent hydrolysis. The glucosinolate-derived aglucones are unstable and rearrange to form isothiocyanates and other products such as the generally less toxic nitriles, either spontaneously or by the action of so-called specifier proteins that help guide rearrangement in *A. thaliana* and other Brassicaceae plants (*Halkier and Gershenzon, 2006*; *Wittstock et al., 2016*; *Wittstock et al., 2004*). Glucosinolate side-chains can further dictate post-hydrolysis reactivities and occasionally lead to additional intramolecular reactions (e.g. cyclization to form oxazolidine-2-thione derivatives) or rapid loss of the –N=C=S group (e.g. to form the carbinol products of indolic glucosinolates) (*Wittstock et al., 2016*), which alter toxicity. Isothiocyanates, the most typical glucosinolate hydrolysis products, are considered toxic to a variety of herbivorous organisms due to their lipophilic properties that facilitate diffusion through membranes and the electrophilic –N=C=S functional core that reacts with intracellular nucleophiles (*Hanschen et al., 2012*;

*Jeschke et al., 2016a*). This liberation of toxic products from glucosinolates constitutes the characteristic 'mustard oil bomb' defense system of plants of the family Brassicaceae and related families.

Some herbivores possess mechanisms to potentially avoid the toxicity of the glucosinolate-myrosinase system (*Jeschke et al., 2016a*; *Winde and Wittstock, 2011*). For example, *Plutella xylostella* (the diamondback moth, Lepidoptera: Plutellidae), a notorious agricultural pest specializing on plants of the Brassicaceae, produces glucosinolate-specific sulfatases that rapidly desulfate glucosinolates to form desulfo-glucosinolates (*Ratzka et al., 2002*), which are no longer substrates for myrosinases and are therefore non-toxic. This process is controlled by a small gene family encoding glucosinolate sulfatases (GSSs) with varying specificity for different types of glucosinolates. GSS1 desulfated all glucosinolates tested in vitro except for 1MOI3M, while GSS3 only metabolized benzenic and indolic glucosinolates and GSS2 accepted only a few very long-chain aliphatic glucosinolates (*Heidel-Fischer et al., 2019*). GSSs may help to avoid the toxicity of the glucosinolate-myrosinase system, and indeed it was reported that *P. xylostella* contains sufficient GSS (based on an in vitro assay of purified protein) to readily desulfate all ingested glucosinolates (*Ratzka et al., 2002*). Moreover, when isotopically labeled glucosinolates were administered to *P. xylostella*, over 80% were converted to desulfo-glucosinolates (*Jeschke et al., 2017*). However, it cannot be automatically assumed that desulfation is an effective detoxification strategy without knowing if it is of net benefit to the performance and fitness of the herbivore, since desulfation may incur unanticipated physiological costs.

*P. xylostella* is currently the most devastating pest of cultivated Brassicaceae crops in the world, causing losses of US\$ 4–5 billion per year (*Zalucki et al., 2012*). Its destructiveness can be attributed to its extremely short life cycle (14 days), which allows it to complete more than 20 generations per year, and its rapid development of resistance to pesticides (*Sarfraz et al., 2005*), making it one of the most difficult agricultural pests to control. 'Natural enemies' have therefore been explored as alternative *P. xylostella* control agents, including predators, parasitoids, entomopathogenic fungi, bacteria and viruses (*Sarfraz et al., 2005*). One natural enemy used in diamondback moth management is the predatory lacewing C*hrysoperla carnea* (the common green lacewing, Neuroptera: Chrysopidae) (*McEwen et al., 2001*; *Reddy et al., 2004*). When *C. carnea* preys on *P. xylostella* larvae feeding on Brassicaceae plants, the lacewing can be expected to encounter plant-derived glucosinolates or their metabolites. However, it is unclear whether such plant defense compounds or derivatives might harm *P. xylostella* predators. Plant chemicals can traverse trophic levels and affect predators, as when they are sequestered by a herbivore for its own defense (*Beran et al., 2014*; *Kumar et al., 2014*; *Müller et al., 2002*). The cabbage aphid, *Brevicoryne brassicae*, for example, accumulates high concentrations of glucosinolates from its host plants and uses these compounds in defense against predators (*Kos et al., 2011*). In this work, we therefore attempted to down-regulate *P. xylostella* GSS activity to determine if glucosinolate desulfation fundamentally benefits *P. xylostella*, and also whether or not it has consequences for a higher trophic level.

We chose plant-mediated RNAi to silence *gss* genes since this method is reported to have high success against lepidopteran targets (*Kumar et al., 2012*; *Kumar et al., 2014*; *Mao et al., 2007*; *Poreddy et al., 2017*). In leaves of the *Arabidopsis thaliana* Col-0 accession used here, aliphatic and indolic glucosinolates constitute around 85% and 15% of the total glucosinolate pool, respectively, with 4-methylsulfinylbutyl glucosinolate (4MSOB) representing over 70% of the aliphatic glucosinolates (*Brown et al., 2003*). Using *A. thaliana* plants with wild-type glucosinolates and *myb28myb29* mutant plants without aliphatic glucosinolates, both engineered to target *gss* gene expression, we achieved significant silencing of *gss* in *P. xylostella*. We demonstrated that suppression of glucosinolate desulfation increased larval isothiocyanate levels, which had significant negative impacts on *P. xylostella* growth, survival and reproduction, establishing that this detoxification mechanism is beneficial to the herbivore in spite of its observed cost. Although the increased isothiocyanate accumulation in *P. xylostella* larvae impaired the growth of predatory *C. carnea* larvae, *C. carnea* detoxified isothiocyanates via the general mercapturic acid pathway and excreted the metabolites in their larval anal secretion resulting in no net effect on pupal mortality and adult egg-laying capacity. Therefore, our work shows that both the herbivore and its predator detoxify plant defensive chemicals with their own independent mechanisms with resulting benefits to reproductive fitness.

## Results

### *gss1* is abundantly expressed in the midgut of *P. xylostella* larvae

To determine the role of glucosinolate sulfatase (GSS) in *P. xylostella* in more detail, we explored the location of *gss* gene expression by qRT-PCR in dissected tissues of fourth-instar larvae. The *gss1* gene was highly expressed in the midgut epithelium, but had very low expression in hemolymph, integument and fat bodies (*Figure 1A*). After feeding on *A. thaliana* wild-type Col-0 plants containing natural levels of glucosinolates, *gss1* expression in larval midgut tissues was approximately 17-fold higher than after feeding on *myb28myb29* plants, which lack aliphatic glucosinolates (*Figure 1A*), suggesting that the expression of this gene is regulated by dietary glucosinolate ingestion. Similar patterns were shown by *gss2* and *gss3* (*Figure 1—figure supplement 1A,B*). However, glucosinolates did not induce the expression of any of several other potential detoxification genes analyzed in the midgut of *P. xylostella* fourth-instar larvae (*Figure 1—figure supplement 1C*).

To analyze the cell-level expression of *gss* within the larval midgut, we conducted fluorescent in situ hybridization (FISH) experiments utilizing antisense *gss*-specific riboprobes. The labeling of *gss1*-expressing cells suggested a broad distribution in cell types, as seen in a typical transverse section of *P. xylostella* midgut (*Figure 1B*). The counter staining of nuclei indicated that *gss1*-positive midgut cell types include the columnar cells bearing microvillar structures, and the basal midgut cells that differ in nuclear shape and size (*Figure 1C,D*). FISH labeling of *gss2* and *gss3* revealed that both have a similar expression pattern as *gss1* (*Figure 1—figure supplement 2A–F*). Moreover two-color FISH experiments using the antisense probe pairs, *gss1* and *gss2* or *gss1* and *gss3*, indicated that the three *gss* forms are most likely co-expressed in the same midgut cells, as inferred by the largely overlapping red and green labeling patterns (*Figure 1—figure supplement 2G–J*). The specificities of the labeling conferred by individual *gss* antisense probes were verified using the corresponding sense probes, which did not generate any labeling (*Figure 1—figure supplement 2K–M*).

### Silencing of *P. xylostella* gss1 expression reduces GSS enzyme activity

To determine the impact of GSS on *P. xylostella* larval performance and glucosinolate metabolism, we used plant-mediated RNAi to downregulate its expression. Transgenic *A. thaliana* lines were generated by infiltration with *Agrobacterium tumefaciens* transformed with a virus-based dsRNA production system, which consisted of the tobacco rattle virus-derived vector (pTRV2) with a 526 bp fragment of *gss1* (*Figure 1E*). Both *A. thaliana* Col-0 and *myb28myb29* plants, with and without aliphatic glucosinolates, respectively, were transformed. As a negative control, plants were also infiltrated with *A. tumefaciens* transformed with an empty vector pTRV2 construct. Plants infiltrated with the *gss1* RNAi construct were indistinguishable from both untreated and empty vector construct-infiltrated plants regarding growth, morphology, glucosinolate profile, and levels of flavonoids and phenylpropanoids involved in plant defense (*D'Auria and Gershenzon, 2005*) (*Figure 1—figure supplement 3*).

When *P. xylostella* larvae ingested Col-0 plants (with wild-type glucosinolate levels) infiltrated with the *gss1* RNAi construct (*Figure 1E*), *gss1* expression in the midgut epithelium was lowered to about 2% of that found in larvae fed on Col-0 plants infiltrated with the empty vector construct (*Figure 1F*). After feeding on *myb28myb29* plants (without aliphatic glucosinolates), larval *gss1* transcript levels were lower than after feeding on Col-0, but *gss1* silencing on *myb28myb29* plants also led to a substantial (63%) transcriptional reduction (*Figure 1F*). Subsequently, we measured the GSS activities in protein extracts from the midgut epithelium of *gss1*-silenced and non-silenced *P. xylostella* larvae using in vitro enzyme assays. Formation of desulfo-4-methylsulfinylbutyl glucosinolate (desulfo-4MSOB) from 4MSOB by midgut extracts of *gss1*-silenced larvae was reduced to less than 50% of that formed by extracts of non-silenced larvae, indicating *gss1* expression and GSS activity were both greatly suppressed by the RNAi treatment (*Figure 1G*). While *gss2* and *gss3*, like *gss1*, were co-silenced by the treatment, transcript levels of genes encoding other sulfatases and the *sulfatase modifying factor 1* (*sumf1*) were not influenced by *gss1* silencing (*Figure 1—figure supplement 4*).

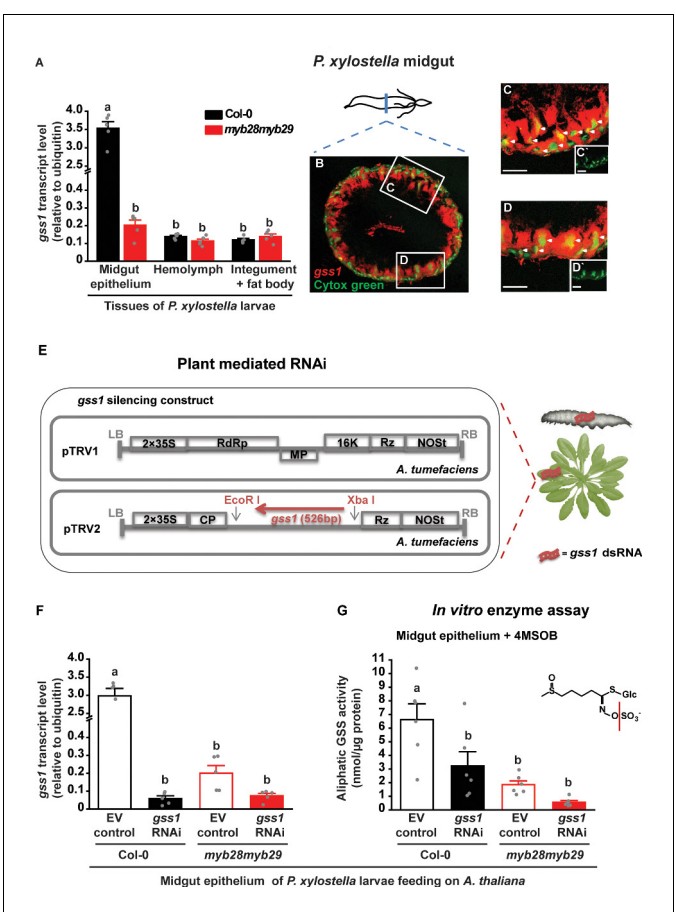

**Figure 1.** Localization and plant-mediated RNAi silencing of *gss1* gene expression in *P. xylostella* larvae. (**A**) Levels of *gss1* transcripts (expressed relative to ubiquitin) are higher in midgut epithelium than in hemolymph, integument and fat bodies of fourth-instar larvae feeding on *A. thaliana* Col-0 (wild-type glucosinolates) and *myb28myb29* (no aliphatic glucosinolates) plants (Plants, $F_{1,24}$ = 338.032, p≤0.0001; Tissues, $F_{2,24}$ = 367.522, p≤0.0001; Plants*Tissues, $F_{2,24}$ = 336.042, p≤0.0001; *n* = 5 for all bars). (**B–D**) Visualization of *gss1*-expressing cells in the midgut of *P. xylostella* as seen in a transverse section. Cells containing *gss1* transcripts were strongly stained by means of fluorescent in situ hybridization (FISH) using a *gss1*-specific antisense riboprobe labeled by digoxigenin (DIG) (in red). Counter staining of nuclei of midgut cells employed Cytox green (in green). Two selected areas in **B** (**C** and **D**) are magnified further. Both the columnar cells and the basal midgut cells are *gss1*-positive, with white arrows pointing to nuclei of different morphologies. **C'** and **D'** are lower magnification views of the nuclear staining presented in **C** and **D**, respectively. Scale bar, 50 μm. (**E**) Silencing strategy for *gss1* employed a virus-based dsRNA-producing system in the host plant *A. thaliana* created by infiltration of tobacco rattle virus (TRV, detail structure described in ***Ratcliff et al., 2001***) engineered to express a 526 bp fragment of *gss1* dsRNA. (**F**) Levels of *gss1* transcripts (expressed relative to ubiquitin) in larval midgut epithelium ($F_{3,16}$ = 185.508, p≤0.0001, *n* = 5 for all bars) and (**G**) levels of GSS activity measured in vitro in extracts of midgut epithelium ($F_{3,20}$ = 10.697, p≤0.0001, *n* = 6 for all bars) of fourth-instar larvae feeding on empty vector (EV) control and *gss1*-RNAi plants in backgrounds of Col-0 and *myb28myb29 A. thaliana*. The aliphatic glucosinolate 4MSOB (5 mM) was used as a substrate. Significant differences (p≤0.05) between means (±s.e.) were determined by Tukey HSD tests in conjunction with a two-way ANOVA in **A** and with one-way ANOVA in **F** and **G**.

The online version of this article includes the following source data and figure supplement(s) for figure 1:

**Source data 1.** *gss1* gene transcript levels and GSS in vitro enzyme assays with 4MSOB glucosinolate.
**Figure supplement 1.** Transcripts of *gss* genes are localized in the *P. xylostella* larval midgut epithelium and are induced by dietary ingestion of glucosinolates.
**Figure supplement 1—source data 1.** Transcript levels of *gss2*, *gss3* and other detoxification-related genes.
**Figure supplement 2.** Transcripts of *gss* genes are localized in midgut cell types of *P. xylostella* fourth-instar larvae.
**Figure supplement 3.** Infiltration of *A.thaliana* Col-0 and *myb28myb29* lines with *gss1* RNAi and empty vector (EV) control constructs does not alter their morphology and chemical phenotypes.
*Figure 1 continued on next page*

*Figure 1 continued*

**Figure supplement 3—source data 1.** Glucosinolate, flavonoid, and phenylpropanoid profiles in *A. thaliana* plants.

**Figure supplement 4.** Plant-mediated RNAi of *gss1* co-silences *gss2* and *gss3* due to high sequence similarity, and suppresses the desulfation of indolic glucosinolates as well.

**Figure supplement 4—source data 1.** *gss2* and *gss3* gene transcript levels, and GSS in vitro enzyme assays with I3M glucosinolate.

## Silencing of *gss* decreases *P. xylostella* growth, survival and reproduction

To determine the impact of GSS on *P. xylostella* performance, we compared the phenotypes of larvae that were either *gss*-silenced or non-silenced and fed on *A. thaliana* plants with or without aliphatic glucosinolates. When fed continuously on Col-0 plants with aliphatic glucosinolates, *gss*-silenced larvae grew 33% less than non-silenced larvae at six dph (days post hatching), and the growth gap continued to widen until eight dph, when *gss*-silenced larvae were only 64% as heavy as non-silenced larvae before pupation (*Figure 2A*). Although pupal weights were not significantly different between treatments (*Figure 2—figure supplement 1*), the pupal mortality of *gss*-silenced insects was nearly 4-fold higher than non-silenced ones (*Figure 2B*).

We then separated the resulting adults by gender to examine the influence of *gss* silencing on fecundity. Female moths arising from the four different larval treatments, *gss*-silenced or non-silenced from either *A. thaliana* Col-0 or *myb28myb29*, were all mated with male moths arising from non-silenced larvae from Col-0, while male moths arising from the same four treatments were all mated with female moths arising from non-silenced larvae from Col-0. The number of eggs laid was counted two days after mating, and the number of eggs hatching successfully was recorded later. Egg laying and hatching were reduced only in treatments involving adults arising from *gss*-silenced larvae fed on aliphatic glucosinolate containing (Col-0) plants. Female moths from *gss*-silenced Col-0 larvae laid 56% less eggs than female moths on all other treatments (*Figure 2C*). However, these eggs did not hatch at a reduced rate (*Figure 2E,G*). Meanwhile, male moths arising from *gss*-silenced larvae fed on Col-0 did not affect the egg laying capacity of females arising from non-silenced larvae (*Figure 2D*), but did decrease egg hatching by 36% (*Figure 2F*). The distribution and lower slopes observed in the correlation between the numbers of eggs hatched and the numbers of eggs laid by this group suggest that the two factors were independent, with low-hatching batches spread among differently sized broods (*Figure 2H*). We confirmed these results by mating adults arising from *gss*-silenced or non-silenced larvae with adults from larvae fed on untreated Col-0 or *myb28myb29* plants to eliminate any influence from infiltration with the empty vector construct (*Figure 2—figure supplement 2*).

Comparing *P. xylostella* performance on the two plant lines, larvae fed on *myb28myb29* plants (without aliphatic glucosinolates) grew faster and had significantly higher mass than larvae fed on Col-0 plants (with aliphatic glucosinolates) (*Figure 2A*). This difference may result from the formation of toxic glucosinolate hydrolysis products despite the presence of GSS. Alternatively, better performance on *myb28myb29* might be ascribed to reduced amounts of glucosinolate sulfatase activity due to decreased expression of the gene (*Figure 1F,G*), indicating a trade-off between energy spent on glucosinolate detoxification and larval growth (*Jeschke et al., 2016b*; *Jeschke et al., 2017*). However, considering only larvae fed on *myb28myb29* plants, *gss* silencing did not cause physiological changes (*Figure 2B–H*). Therefore, silencing of *gss* impacted *P. xylostella* larval growth and development only in the presence of aliphatic glucosinolates, causing pupal mortality and sex-specific effects on adults.

## Silencing of *gss* increases the formation of isothiocyanates, toxic glucosinolate hydrolysis products, in *P. xylostella* larvae

To determine the influence of *gss* silencing on insect glucosinolate metabolism after ingestion, we quantified the previously described (*Jeschke et al., 2017*) metabolites of 4-methylsulfinylbutyl glucosinolate (4MSOB), which represents 75% of the total aliphatic glucosinolate content of *A. thaliana* Col-0 leaves (*Figure 1—figure supplement 3*). Non-silenced fourth-instar larvae with normal levels

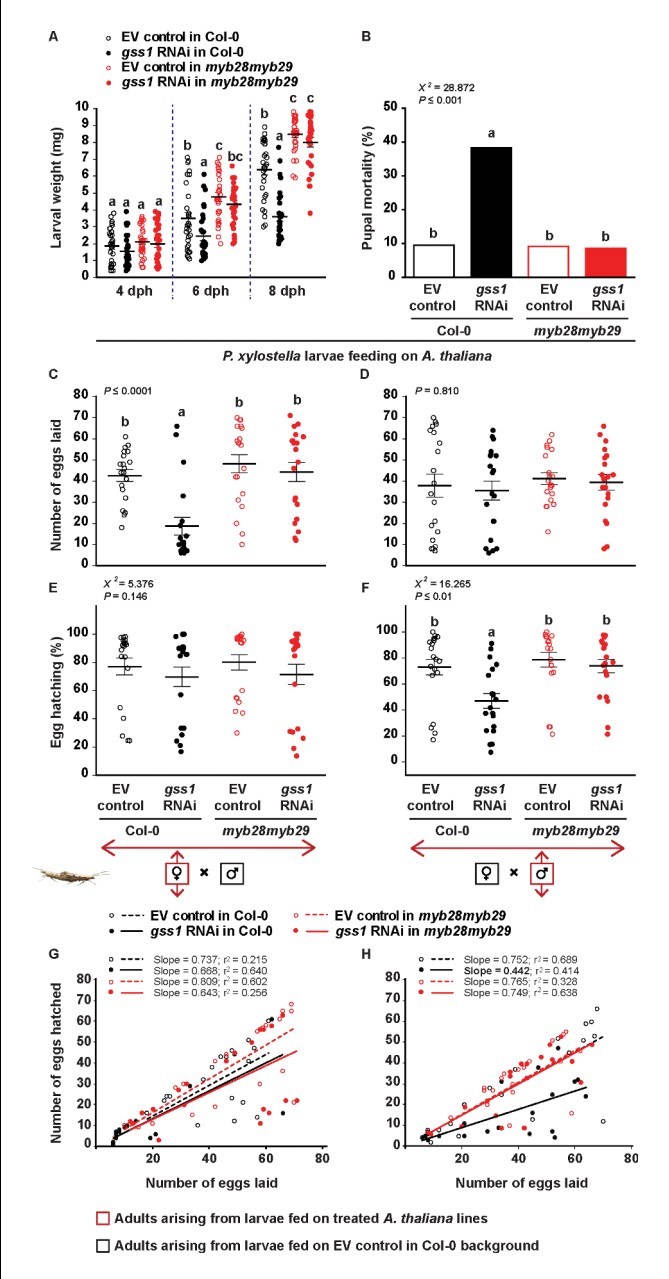

**Figure 2.** *Silencing of gss affects P. xylostella growth and fitness in the presence of aliphatic glucosinolates. Larvae were fed on empty vector (EV) control and gss1 RNAi plants from either Col-0 (wild-type glucosinolates) or myb28myb29 (no aliphatic glucosinolates) backgrounds, and various parameters of larval performance were measured:* (A) *Weights of larvae after four dph (days post hatching), six dph, and eight dph were reduced by silencing but only in the presence of wild-type glucosinolates (six dph, Plants $F_{1,116}$ = 33.471, p≤0.0001; RNAi $F_{1,116}$ = 7.440, p≤0.01; Plants\*RNAi $F_{1,116}$ = 1.322, p=0.253. eight dph, Plants $F_{1,116}$ = 145.33, p≤0.0001; RNAi $F_{1,116}$ = 37.083, p≤0.0001; Plants\*RNAi $F_{1,116}$ = 18.340, p≤0.0001. n = 30 for all treatments);* (B) *Mortality (%) in pupae was highest in those pupae arising from silenced larvae fed on wild-type glucosinolates ($\chi^2$ = 228.872, df = 3, p≤0.001; n = 63, 60, 66 and 59, respectively). The number of eggs laid by females was measured in crosses between* (C) *females arising from larvae raised on the four plant lines mated with males arising from larvae fed on EV control Col-0 plants and* (D) *males arising from larvae raised on the four plant lines mated with females arising from larvae fed on EV control Col-0 plants. Only crosses with females arising from gss-silenced larvae fed on Col-0 showed a decline in egg production ($F_{3,76}$ = 11.157, p≤0.0001, n = 20 for all treatments). Hatching (%) of eggs laid by females* (E, F), *and the correlation between the numbers of eggs hatched and the numbers of eggs laid* (G, H) *were recorded from crosses between* (E, G) *females arising from larvae raised on the four plant lines mated with*

*Figure 2 continued on next page*

*Figure 2 continued*

males arising from larvae fed on EV control Col-0 plants, and (**F, H**) from males arising from larvae raised on the four plant lines mated with females arising from larvae fed on EV control Col-0 plants. Only crosses with males arising from *gss*-silenced larvae fed on Col-0 showed a decline in egg hatching rate ($\chi^2$ = 216.265, *df* = 3, p≤0.01; n = 20 for all treatments). Significant differences (p≤0.05) between means (±s.e.) were determined by Tukey HSD tests in conjunction with two-way ANOVA in **A**, a proportions test with pairwise comparisons in **B**, Tukey HSD tests in conjunction with one-way ANOVA in **C–D**, and Dunn's post hoc tests in conjunction with non-parametric Kruskal-Wallis tests in **E–F**.

The online version of this article includes the following source data and figure supplement(s) for figure 2:

**Source data 1.** *P. xylostella* larval weight, pupal mortality, and adult egg laying capacity.
**Figure supplement 1.** Silencing of *gss* has no effect on weights of *P. xylostella* pupae.
**Figure supplement 1—source data 1.** *P. xylostella* pupal weight.
**Figure supplement 2.** Silencing of *gss* reduces number of eggs and their hatching percentage.
**Figure supplement 2—source data 1.** *P. xylostella* adult egg laying capacity.

---

of GSS activity desulfated 4MSOB and formed the non-toxic desulfo-4MSOB in the larval midgut epithelium (*Figure 3B*). However, when GSS activity was lowered by plant-mediated RNAi, levels of desulfo-4MSOB decreased by more than 90%, while the concentrations of the toxic hydrolysis product 4MSOB-isothiocyanate (4MSOB-ITC) increased over 10-fold in the midgut of *gss*-silenced larvae, with a concomitant rise in the hemolymph (*Figure 3C*). Intact 4MSOB, present only in trace amounts in non-silenced midguts, became a very prominent peak in *gss*-silenced larvae (*Figure 3A*), suggesting that plant myrosinases in the *P. xylostella* midgut lumen are not very efficient, and so may be easily outcompeted by insect GSS in non-silenced insects. Larvae of *P. xylostella* excreted most of the 4MSOB and its metabolites into the frass. Active GSSs and myrosinase were also excreted that continued to react with the 4MSOB present (*Figure 3—figure supplement 1*), likely leading to the smaller differences in desulfo-4MSOB and 4MSOB-ITC concentrations between the frass of *gss*-silenced larvae and non-silenced insects as between the concentrations of these compounds in the midgut (*Figure 3B,C*). At the end of larval development, *P. xylostella* retained considerable amounts of 4MSOB-ITC in pupae and adults, 210 and 390 nmol/g, respectively (*Figure 3C*). However, the desulfo-4MSOB remaining was excreted at the prepupal stage (*Figure 3B*). Neither desulfo-4MSOB nor 4MSOB-ITC were detectable in *P. xylostella* eggs (*Figure 3B,C*).

The desulfation of indol-3-ylmethyl (I3M) glucosinolate was also inhibited in *gss*-silenced *P. xylostella* larvae. This glucosinolate, which represents 76% and 87% of the indolic glucosinolate pool in *A. thaliana* Col-0 and *myb28myb29*, respectively (*Figure 1—figure supplement 3B*), was shown to be less efficiently desulfated in *gss*-silenced than empty vector control lines according to in vitro enzyme assays (*Figure 1—figure supplement 4*) and metabolomic analyses (*Figure 3—figure supplement 2*). Nevertheless, the lack of indolic glucosinolate desulfation had no observable negative effects on the performance of *P. xylostella* larvae (*Figure 2*). There were no differences between silenced and non-silenced larvae feeding on *myb28myb29* plants, which lack aliphatic glucosinolates and contain slightly elevated indolic glucosinolate levels (*Figure 1—figure supplement 3B*). Thus the metabolites of I3M appear to be less toxic than those of 4MSOB to *P. xylostella*, as also observed for other herbivores (*Jeschke et al., 2017*). In addition to the lower amounts of I3M than 4MSOB in *A. thaliana*, the I3M-isothiocyanate (I3M-ITC) formed upon hydrolysis is unstable under physiological conditions and reacts largely with ascorbic acid (*Figure 3—figure supplement 2*).

## 4MSOB-ITC is responsible for the negative effects of *gss* silencing on *P. xylostella* performance

As isothiocyanates are thought to cause most of the toxic effects of glucosinolates (*Brown and Hampton, 2011*; *Wittstock et al., 2003*), we hypothesized that the significantly higher concentrations of 4MSOB-ITC resulting from *gss* silencing would explain the lower larval growth, higher pupal mortality, and reduced reproduction. To examine this possibility, *myb28myb29* plants lacking 4MSOB and transformed with either the *gss1* RNAi or empty vector constructs were additionally infiltrated with a natural concentration of 4MSOB-ITC in a solvent of 0.4% aqueous ethanol (*Figure 4—figure supplement 1*). Solvent-infiltrated *myb28myb29* plants were used as negative controls. Feeding on 4MSOB-ITC-infiltrated plants strongly reduced larval growth (*Figure 4A*), and

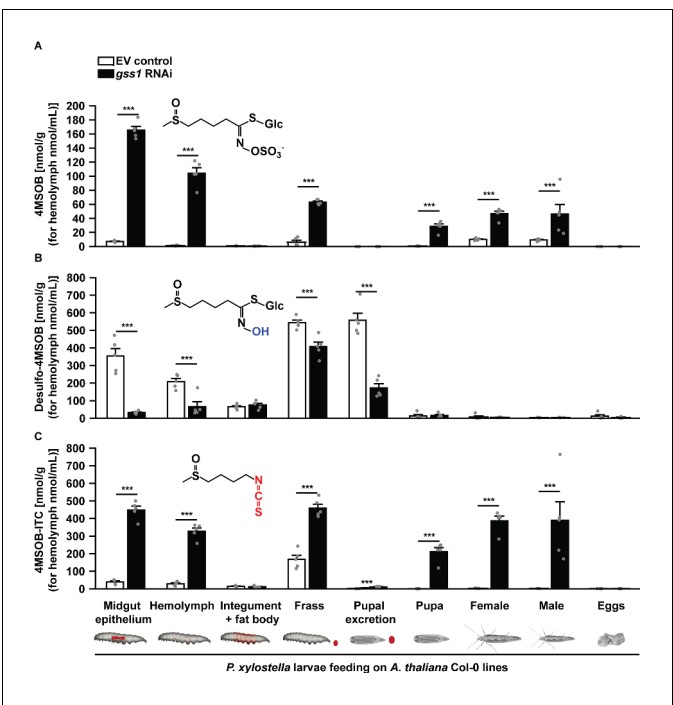

**Figure 3.** Silencing of *gss* decreases desulfo-4MSOB and increases 4MSOB-ITC concentrations in *P. xylostella*. (**A**) Intact 4MSOB (Midgut epithelium, $F_{3,16} = 931.79$, $p \leq 0.0001$; Hemolymph, $F_{3,16} = 172.90$, $p \leq 0.0001$; Frass, $F_{3,16} = 391.817$, $p \leq 0.0001$; Pupa, $F_{3,16} = 68.59$, $p \leq 0.0001$; Female, $F_{3,16} = 150.48$, $p \leq 0.0001$; Male, $F_{3,16} = 9.64$, $p \leq 0.001$; $n = 5$ for all bars), (**B**) desulfo-4MSOB (Midgut epithelium, $F_{3,16} = 66.94$, $p \leq 0.0001$; Hemolymph, $F_{3,16} = 34.84$, $p \leq 0.0001$; Frass, $F_{3,16} = 329.97$, $p \leq 0.0001$; Pupal excretion, $F_{3,16} = 129.825$, $p \leq 0.0001$; $n = 5$ for all bars), and (**C**) 4MSOB-ITC (Midgut epithelium, $F_{3,16} = 356.30$, $p \leq 0.0001$; Hemolymph, $F_{3,16} = 269.44$, $p \leq 0.0001$; Frass, $F_{3,16} = 182.96$, $p \leq 0.0001$; Pupal excretion, $F_{3,16} = 81.248$, $p \leq 0.0001$; Pupa, $F_{3,16} = 78.44$, $p \leq 0.0001$; Female, $F_{3,16} = 211.63$, $p \leq 0.0001$; Male, $F_{3,16} = 13.83$, $p \leq 0.0001$; $n = 5$ for all bars) were measured in various larval tissues, excretions, pupae, and adult moths developed from *gss*-silenced and non-silenced larvae feeding on *A. thaliana* Col-0 (wild-type glucosinolates). The concentrations of 4MSOB and its metabolites were nearly undetectable in *P. xylostella* fed on *myb28myb29* (no aliphatic glucosinolates) plants, and these data are not shown in the figure, but given in *Figure 3—source data 1*. Significant differences ($p \leq 0.05$) between means (±s.e.) were determined by Tukey HSD tests in conjunction with one-way ANOVA.

The online version of this article includes the following source data and figure supplement(s) for figure 3:

**Source data 1.** The concentration of 4MSOB glucosinolate metabolites in *P. xylostella*.
**Figure supplement 1.** Enzymatic conversion of 4MSOB to desulfo-4MSOB and 4MSOB-ITC is enzymatic and occurs in fresh frass.
**Figure supplement 1—source data 1.** Enzyme assay of *P. xylostella* larval frass with 4MSOB glucosinolate.
**Figure supplement 2.** Silencing of *gss* decreases desulfo-indol-3-ylmethyl glucosinolate (desulfo-I3M) and increases a metabolite of indol-3-ylmethyl glucosinolate isothiocyanate (I3M-ITC) in *P. xylostella*.
**Figure supplement 2—source data 1.** The concentration of I3M glucosinolate metabolites in *P. xylostella*.

pupae of larvae fed on 4MSOB-ITC-infiltrated plants suffered 4-fold higher mortality than those feeding on solvent-infiltrated plants (*Figure 4B*). Furthermore, the female moths of larvae feeding on 4MSOB-ITC-infiltrated plants laid approximately 45% less eggs than those feeding on solvent-infiltrated plants (*Figure 4C*), and eggs fertilized by male moths of larvae feeding on 4MSOB-ITC-infiltrated plants had approximately 50% lower hatching success (*Figure 4F,H*). Therefore, the negative physiological effects suffered by *gss*-silenced *P. xylostella* larvae fed on *A. thaliana* Col-0 plants with aliphatic glucosinolates are likely caused by the exposure to 4MSOB-ITC resulting from hydrolysis of glucosinolates that were not efficiently desulfated as in non-silenced insects.

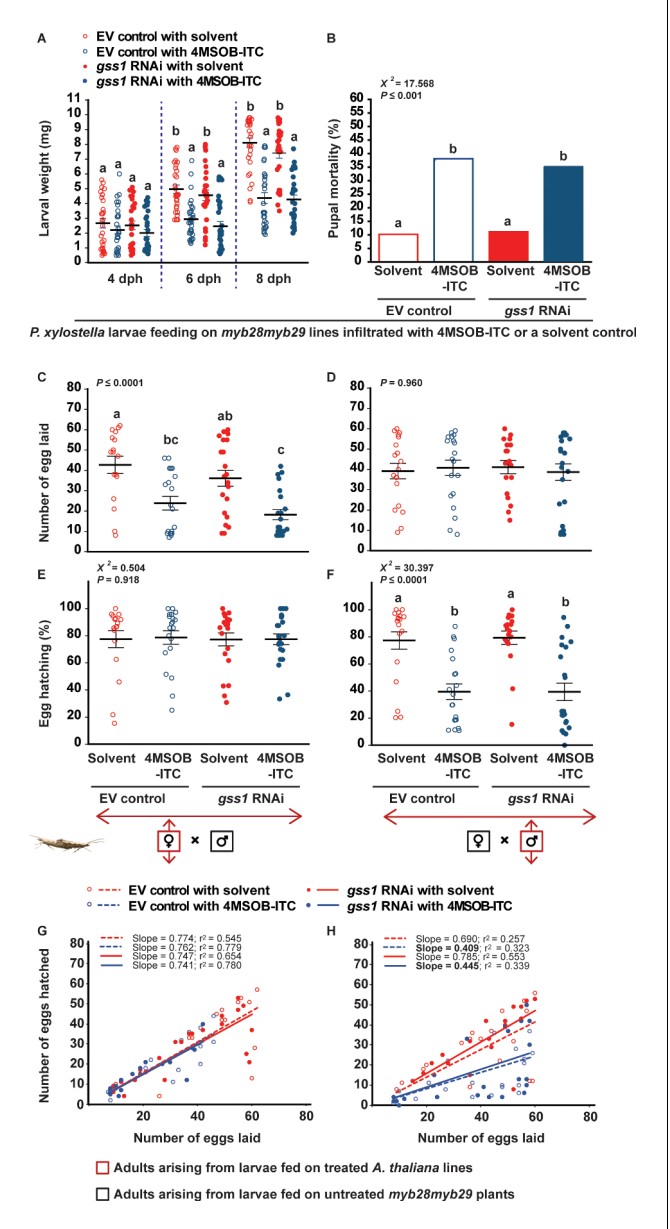

**Figure 4.** Negative effects on *P. xylostella* fitness after *gss* silencing are attributed to the increased 4MSOB-ITC concentrations in the insect body. Larvae that were *gss*-silenced and non-silenced were fed on *myb28myb29* leaves infiltrated with 4MSOB-ITC (dissolved in a 0.4% aqueous ethanol solvent) or solvent-infiltrated control leaves and the following parameters were measured: (**A**) weights of larvae four dph (days post hatching), six dph and eight dph, (**B**) mortality (%) in pupae formed from treated larvae, (**C**) number of eggs laid by females arising from treated larvae mated with males developed from control larvae feeding on untreated *myb28myb29* plants, (**D**) number of eggs laid by female control larvae mated with males arising from treated larvae, (**E**) hatching (%) of eggs laid by females arising from treated larvae mated with males developed from control larvae, (**F**) hatching (%) of eggs laid by females arising from control larvae mated with males developed from treated larvae, and the correlations between the numbers of eggs hatched and the numbers of eggs laid (**G**) by females arising from treated larvae mated with males arising from control larvae, and (**H**) by females arising from control larvae mated with males arising from treated larvae. Feeding on 4MSOB-ITC caused a significant decline in larval weight (six dph, RNAi $F_{1,116}$ = 2.130, p=0.147; Infiltration $F_{1,116}$ = 46.287, p≤0.0001; RNAi*Infiltration $F_{1,116}$ = 0.012, p=0.913. eight dph, RNAi $F_{1,116}$ = 1.394, p=0.240; Infiltration $F_{1,116}$ = 103.860, p≤0.0001; RNAi*Infiltration $F_{1,116}$ = 0.765, p=0.384. n = 30 for all treatments), a significant increase in pupal mortality ($\chi^2$ = 217.568, df = 3, p≤0.001; n = 49, 42, 45 and 51, respectively), a significant decline in egg laying in crosses when females arise from larvae fed on 4MSOB-ITC ($F_{3,76}$ = 10.046, p≤0.0001, n = 17, 20, 21 and 22, respectively), and a significant decline in egg

*Figure 4 continued on next page*

*Figure 4 continued*

hatching rate when males arose from larvae fed on 4MSOB-ITC ($\chi^2$ = 230.397, *df* = 3, p≤0.0001; *n* = 19, 20, 18 and 22, respectively). Significant differences (p≤0.05) between means (±s.e.) were determined by Tukey HSD tests in conjunction with two-way ANOVA in **A**, a proportions test with pairwise comparisons in **B**, Tukey HSD tests in conjunction with one-way ANOVA in **C–D**, and Dunn's post hoc tests in conjunction with non-parametric Kruskal-Wallis tests in **E–F**.

The online version of this article includes the following source data and figure supplement(s) for figure 4:

**Source data 1.** *P. xylostella* larval weight, pupal mortality, and adult egg laying capacity in complementation experiment.

**Figure supplement 1.** Concentrations of 4MSOB-ITC and conjugates formed in crushed leaves of Col-0 *A. thaliana* plants, and in *myb28myb29* leaves infiltrated with 4MSOB-ITC in aqueous ethanol or with pure aqueous ethanol as a control.

**Figure supplement 1—source data 1.** The concentration of 4MSOB-ITC derivates in *A. thaliana* plants.

## Lacewing larvae predating upon *gss*-silenced *P. xylostella* detoxify and mobilize activated glucosinolate hydrolysis products

When consuming *P. xylostella* larvae or other herbivores feeding on glucosinolate-containing plants, predators such as the lacewing *C. carnea* almost inevitably encounter glucosinolates or their metabolic products. To determine the effect of these compounds on a herbivore predator, we therefore explored the consequences of altering *P. xylostella* larval glucosinolate metabolism on *C. carnea*. When larvae of this lacewing preyed on *gss*-silenced *P. xylostella* larvae fed on plants containing aliphatic glucosinolates, significantly greater amounts of 4MSOB-ITC (and significantly lesser amounts of desulfo-4MSOB) deriving from *P. xylostella* larvae were present in the gut, hemolymph and Malpighian tubules compared to *C. carnea* preying on non-silenced *P. xylostella* larvae (*Figure 5B,C*). The lacewing metabolized a large portion of the 4MSOB-ITC ingested from *gss*-silenced *P. xylostella* larvae via the general mercapturic acid pathway (*Figure 5A*), an isothiocyanate detoxification pathway used by many organisms (*Gloss et al., 2014*; *Schramm et al., 2012*), resulting in 44 nmol/mL and 67 nmol/g of the *N*-acetylcysteine conjugate of 4MSOB-ITC (4MSOB-ITC-NAC) in the hemolymph and Malpighian tubules, respectively (*Figure 5D* and *Figure 5—figure supplement 1*). Due to the lack of connection between midguts and hindguts of *C. carnea* larvae (*Figure 5E*) (*McEwen et al., 2001*), the soluble contents of the midgut, including glucosinolates and their metabolites, can only be excreted after being taken up into the hemolymph, secreted into the Malpighian tubules, transported to the silk-separating reservoir, and deposited in the anal secretion (*Figure 5E*) (*McEwen et al., 2001*). For *C. carnea* preying on *gss*-silenced *P. xylostella*, 327 nmol/mL 4MSOB-ITC and 116 nmol/mL of the detoxification product, 4MSOB-ITC-NAC, were detected in the anal secretion (*Figure 5C,D*). The low amount of 4MSOB-ITC remaining in *C. carnea* larvae was excreted in the pupal pellet during pupation, resulting in undetectable 4MSOB-ITC residues in adults (*Figure 5C*). Lacewing larvae preying on non-silenced *P. xylostella* larvae feeding on Col-0 plants with aliphatic glucosinolates excreted approximately 320 nmol/g of desulfo-4MSOB in the pupal pellet, resulting in virtually no detectable desulfo-glucosinolate being retained in adults and the meconium (*Figure 5B*).

## Ingestion of prey-derived isothiocyanates slows lacewing development, but has no effects on adult fitness and prey choice

The metabolism of toxic glucosinolate hydrolysis products by *C. carnea* incurs costs that are visible as a slight delay in its larval development. Lacewing larval growth was reduced when feeding on *gss*-silenced isothiocyanate-containing *P. xylostella* larvae fed on Col-0 plants (with wild-type glucosinolates) compared to non-silenced *P. xylostella* larvae or *P. xylostella* larvae fed on *myb28myb29* plants (without aliphatic glucosinolates) (*Figure 6A*). Additionally, pupation was delayed by about two days for *C. carnea* larvae fed on the *gss*-silenced *P. xylostella* larvae raised on Col-0 plants, but without affecting the final proportion of larvae successfully reaching the pupal stage (*Figure 6B* and *Figure 6—source data 1*). Similarly, the exposure to isothiocyanates from *gss*-silenced prey also did not affect *C. carnea* pupal mortality and adult egg-laying capacity (*Figure 6C,D*). Thus, ingestion of *gss*-silenced *P. xylostella* larvae with higher 4MSOB-ITC concentrations only caused a slight delay in

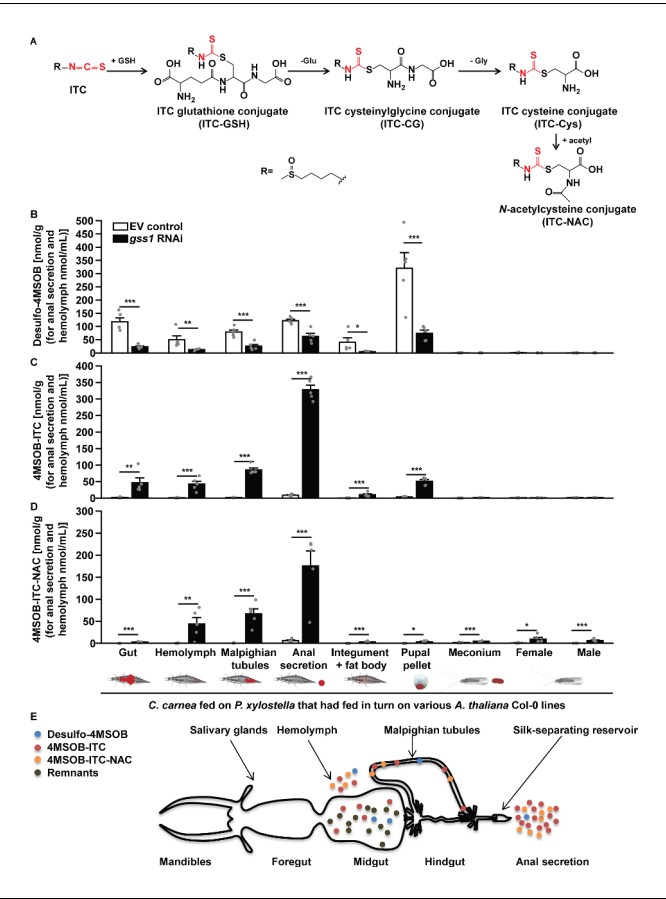

**Figure 5.** The lacewing *C. carnea* circumvents the toxicity of glucosinolate hydrolysis products by conjugation and mobilization. (**A**) General mercapturic acid pathway for detoxification of 4MSOB-ITC in various insects: ingested 4MSOB-ITC is detoxified by conjugation with glutathione (GSH), followed by cleavage to recover glutamate (Glu) and glycine (Gly) and further modification of cysteine (Cys) to form the final *N*-acetylcysteine conjugate (4MSOB-ITC-NAC). Distribution of (**B**) desulfo-4MSOB, (**C**) 4MSOB-ITC, and (**D**) 4MSOB-ITC-NAC conjugate in various tissues and excretions of *C. carnea* larvae and adults arising from larvae that predated upon *gss*-silenced and non-silenced *P. xylostella* larvae feeding on *A. thaliana* Col-0 (wild-type glucosinolates) and *myb28myb29* (no aliphatic glucosinolates) plants. Predation on *gss*-silenced *P. xylostella* larvae caused a significant reduction in desulfated 4MSOB (Gut, $F_{3,16}$ = 32.378, p≤0.0001; Hemolymph, $F_{3,16}$ = 10.23, p≤0.001; Malpighian tubules, $F_{3,16}$ = 50.855, p≤0.0001; Anal secretion, $F_{3,16}$ = 199.006, p≤0.0001; Integument and fat body, $F_{3,16}$ = 5.959, p≤0.01; Pupal pellet, $F_{3,16}$ = 24.907, p≤0.001; n = 5 for all bars), a significant increase in the toxic hydrolysis product, 4MSOB-ITC (Gut, $F_{3,16}$ = 9.895, p≤0.001; Hemolymph, $F_{3,16}$ = 22.967, p≤0.0001; Malpighian tubules, $F_{3,16}$ = 180.333, p≤0.0001; Anal secretion, $F_{3,16}$ = 110.697, p≤0.0001; Integument and fat body, $F_{3,16}$ = 13.919, p≤0.001; Pupal pellet, $F_{3,16}$ = 14.571, p≤0.0001; n = 5 for all bars), and a significant increase in the detoxification product 4MSOB-ITC-NAC (Gut, $F_{3,16}$ = 29.098, p≤0.0001; Hemolymph, $F_{3,16}$ = 9.242, p≤0.001; Malpighian tubules, $F_{3,16}$ = 33.688, p≤0.0001; Anal secretion, $F_{3,16}$ = 32.659, p≤0.0001; Integument and fat body, $F_{3,16}$ = 14.981, p≤0.0001; Pupal pellet, $F_{3,16}$ = 6.544, p≤0.01; Meconium, $F_{3,16}$ = 18.232, p≤0.0001; Female, $F_{3,16}$ = 5.547, p≤0.01; Male, $F_{3,16}$ = 16.777, p≤0.0001; n = 5 for all bars). Since the concentrations of 4MSOB and its metabolites were nearly undetectable in *C. carnea* preying on *P. xylostella* whose larvae fed on *myb28myb29* plants, these data are not shown in the figure, but given in *Figure 5—source data 1*. (**E**) A schematic representation of the alimentary tract of a *C. carnea* larva preying on *P. xylostella* *gss*-silenced larvae feeding on Col-0 (redrawn from *McEwen et al., 2001*) showing storage of the majority of free and conjugated isothiocyanates and derivatives in the anal secretion reservoir. Significant differences (p≤0.05) between means (±s.e.) were determined by Tukey HSD tests in conjunction with one-way ANOVA in **B–D**).

The online version of this article includes the following source data and figure supplement(s) for figure 5:

**Source data 1.** The concentration of 4MSOB glucosinolate metabolites in *C. carnea*.
**Figure supplement 1.** *C. carnea* detoxifies 4MSOB-ITC by forming the 4MSOB-ITC-NAC conjugate.
**Figure supplement 1—source data 1.** The concentration of 4MSOB-ITC derivates in third-trophic level.

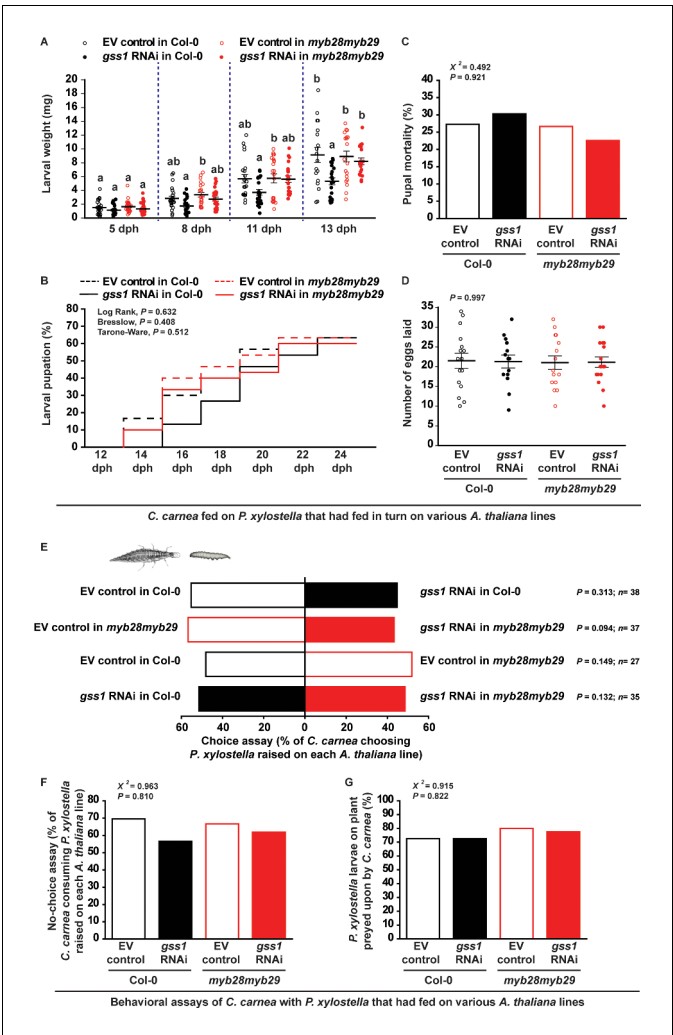

**Figure 6.** *C. carnea* larval development is slowed by glucosinolate metabolites in its prey, but behavior and adult fitness are not affected. (**A**) *C. carnea* larval weights (eight dph, days post hatching, $F_{3,90}$ = 5.164, p≤0.05, *n* = 24, 23, 23 and 24, respectively; eleven dph, $F_{3,82}$ = 3.235, p≤0.05, *n* = 22, 21, 22 and 21, respectively; thirteen dph, $F_{3,73}$ = 5.767, p≤0.01, *n* = 18, 20, 20 and 19, respectively) were decreased when predating on *gss*-silenced *P. xylostella* larvae feeding on *A. thaliana* Col-0 with wild-type glucosinolate levels in comparison to predation on non-silenced larvae or larvae feeding on plants without aliphatic glucosinolates. (**B**) Although the duration of the pupal stage (EV control in Col-0: 18.3 day; *gss1* RNAi in Col-0: 20.9 day; EV control in *myb28myb29*: 18.6 day; *gss1* RNAi in *myb28myb29*: 19.1 day; see *Figure 6—source data 1*) was influenced by the prey, the final percentages of pupation between 12 to 24 dph (Log Rank, *df* = 3, p=0.632; Breslow, *df* = 3, p=0.408; Tarone-Ware, *df* = 3, p=0.512; *n* = 30 for all treatments) were not affected. (**C**) *C. carnea* pupal mortality (%) ($\chi^2$ = 20.492, *df* = 3, p=0.921; *n* = 33, 33, 30 and 31, respectively) and (**D**) adult egg-laying capacity ($F_{3,58}$ = 0.018, p=0.997, *n* = 16, 14, 15, 17, respectively) were not significantly affected by predation upon *gss*-silenced and non-silenced *P. xylostella* larvae feeding on Col-0 (wild-type glucosinolates) or *myb28myb29* (no aliphatic glucosinolates) plants. (**E**) Choice assays, (**F**) no-choice assays ($\chi^2$ = 20.963, *df* = 3, p=0.810; *n* = 23, 23, 21 and 21, respectively), and (**G**) predation trials on plants ($\chi^2$ = 20.915, *df* = 3, p=0.822; *n* = 40 in all bars) showed that *C. carnea* did not avoid *gss*-silenced *P. xylostella* larvae, in spite of their higher 4MSOB-ITC concentrations. Significant differences (p≤0.05) between means (±s.e.) were determined by Tukey HSD tests in conjunction with one-way ANOVA in **A,D**, Kaplan-Meier survival analysis test in **B**, two-sided proportions test in **C,F,G**, and *C. carnea* prey choices were analyzed with a two-sided binomial test (between two *P. xylostella* larvae) and GLM with a binomial distribution and a logit link function (between each treatment) in **E**.

The online version of this article includes the following source data for figure 6:

**Source data 1.** *C. carnea* larval weight, the percentages of pupation and pupal mortality, adult egg laying capacity, and behavior assays.

larval growth, but did not reduce pupal or adult performance compared to consumption of non-silenced *P. xylostella* larvae or *gss*-silenced *P. xylostella* fed on *myb28myb29* plants (*Figure 6A–D*). The ability of *C. carnea* to detoxify isothiocyanates is also reflected in the fact that its behavior is not much influenced by the glucosinolate processing capacity of its prey. This lacewing displayed no particular feeding preference towards either *gss*-silenced or non-silenced larvae fed on either Col-0 or *myb28myb29* plants (*Figure 6E,F*). Similarly, the percentages of *P. xylostella* larvae from each treatment preyed upon by *C. carnea* were nearly identical (*Figure 6G*).

## Discussion

### The GSS system is induced by glucosinolate ingestion: trade-off between development and detoxification

Herbivorous insects have developed a variety of mechanisms to overcome plant defense compounds (*Després et al., 2007*), among which detoxification involving functionally diverse enzymes is most often studied (*Heidel-Fischer and Vogel, 2015*). In several cases, expression of detoxification-related genes and resulting enzyme activities are induced in the herbivore midgut in response to plant defense compounds (*Adesanya et al., 2017*; *Kumar et al., 2014*; *Schweizer et al., 2017*). We show that this scenario also holds true for the diamondback moth (*Plutella xylostella*), a herbivore specialized on plants of the Brassicaceae that employs a glucosinolate sulfatase (GSS) activity to disarm the glucosinolate-myrosinase defense system of its host plants (*Ratzka et al., 2002*). Although GSS had been assumed to be constitutively present in *P. xylostella* (*Winde and Wittstock, 2011*), by comparing larvae feeding on *A. thaliana* plants differing in glucosinolate contents, we and others (*Heidel-Fischer et al., 2019*) have observed the induction of *gss* transcript levels and GSS enzyme activity in the midgut upon dietary glucosinolate ingestion (*Figure 1A,G*). The GSS detoxification system of the specialized herbivore *P. xylostella* is therefore also inducible, similarly to the GSS activity previously identified in the generalist-feeding desert locust (*Schistocerca gregaria*) (*Falk and Gershenzon, 2007*).

The inducibility of GSS can be seen as a way for the herbivore to minimize the physiological costs of detoxification. By producing this enzyme only when needed and predominantly in the gut epithelial tissues (*Figure 1B*) that are the first cells to take up glucosinolates from the gut, the insect may minimize the resources needed for GSS synthesis and maintenance. However, induced GSS detoxification still imposes costs, as inferred from slower larval development when feeding on glucosinolate-containing wild-type plants relative to aliphatic glucosinolate-depleted mutant plants (*Figure 2A*). A trade-off between growth and detoxification capacity has also been noted for other insect herbivores feeding on glucosinolate-containing plants. Several generalist lepidopteran caterpillars resist glucosinolate poisoning by metabolizing isothiocyanates, the chief toxic products of glucosinolate hydrolysis, to their corresponding glutathione (GSH) conjugates. However, the cost of biosynthesizing the GSH necessary for isothiocyanate detoxification leads to delayed larval development (*Jeschke et al., 2016b*; *Jeschke et al., 2017*). Our findings on the plasticity of *gss* gene transcription in *P. xylostella* larvae and its trade-off with growth are consistent with these results. Further research is needed to understand the regulatory machinery behind growth-detoxification trade-offs.

### Plant-mediated RNAi efficiently silences *P. xylostella gss* with severe physiological and fitness consequences

In order to investigate the importance of GSS on *P. xylostella* performance and glucosinolate metabolism, we attempted to reduce GSS activity by *gss* gene silencing. Silencing the expression of target genes by RNAi is often used to clarify their functions in vivo (*Poreddy et al., 2015*; *Wei et al., 2019*; *Zhang et al., 2015*), and is gaining increasing interest as a tool for crop protection against insect pests (*Zhang et al., 2017*; *Zotti et al., 2018*). Nevertheless, it has been documented that RNAi is less efficient in lepidopterans than in other insect orders and gives rise to varying effects in different species (*Shukla et al., 2016*; *Terenius et al., 2011*). Previous studies on *Heliothis virescens* revealed that exogenous dsRNA is not only readily degraded, but also lacks intracellular transport in larvae, reducing the efficiency of RNAi (*Shukla et al., 2016*). *P. xylostella*, on the other hand, has been successfully silenced by feeding droplets of dsRNA (*Bautista et al., 2009*) and bacterially produced dsRNA (*Israni and Rajam, 2017*). However, as a result of the absence of RNA-dependent RNA

polymerases in insects, continuous feeding is required to maintain gene silencing (*Scott et al., 2013*). Plant-mediated dsRNA delivery has been shown to be an effective way to silence target genes in lepidopterans (*Kumar et al., 2014*; *Mao et al., 2007*; *Poreddy et al., 2017*; *Poreddy et al., 2015*; *Zhang et al., 2017*), for which dsRNA can be constantly incorporated by dietary ingestion. In our research, qPCR and in vitro enzyme assays showed that all three *gss* genes were significantly silenced by plant-mediated RNAi in *P. xylostella* (*Figure 1F* and *Figure 1—figure supplement 4A*), while other sulfatases were not affected (*Figure 1—figure supplement 4C*).

Silencing of *gss* in *P. xylostella* larvae severely reduced the enzymatic efficiency of glucosinolate desulfation as measured by both in vitro assays of midgut protein extracts (*Figure 1G*) and glucosinolate metabolite profiles (*Figure 3*). This was accompanied by a large increase in isothiocyanates (*Figure 3C*), the typical toxic products arising from hydrolysis of aliphatic glucosinolates. These metabolic changes resulted in a dramatic decline in larval growth (50%) and an increase in pupal mortality (4-fold) (*Figure 2A,B*), as well as a decline in both egg laying (over 50%) and in egg hatching (about 30%) in different mating combinations (*Figure 2C,F*). The changes observed in egg laying and hatching indicate a sex-specific effect of ITCs, wherein ITC-exposed females laid fewer eggs, and mating with ITC-exposed males decreased the success of egg hatching (*Figure 2C,F*). Interestingly, egg clutches seem to gather into two separate clusters of high and low hatching success. Hatching success was not linked to the numbers of eggs laid by each female, with low-hatching egg batches distributed among clutches of different sizes (*Figure 2G,H*). We can also discard effects from parental ages as a source of this distribution, as only newly emerged moths were paired; however, the frequency and duration of copulation were not recorded. Bimodal egg hatching patterns were also observed when larvae fed on infiltrated ITC (*Figure 4*) and on *myb28myb29* plants (*Figure 2—figure supplement 2*) instead of Col-0 EV controls (*Figure 2*), indicating this effect was consistent among experiments and suggesting it is linked to mating under these conditions. For instance, the isolation of mating pairs into one couple per mating chamber, preventing females from being able to choose among mates, could have led to incompatible matings. It has been documented that repeated mating and multiple partners increase insect egg fertility, with females able to invest less in incompatible males or entirely avoiding them, while incompatible pairings can lead to less viable or unfertilized eggs being laid (*Arnqvist and Nilsson, 2000*; *Saheb et al., 2009*; *Tregenza and Wedell, 2002*). However, this bimodality and the sex-specific effects of ITCs on *P. xylostella* reproduction deserve further investigation.

The negative effects of *gss* silencing on *P. xylostella* growth and reproduction were also observed when larvae fed on leaves from glucosinolate-lacking mutant plants that were complemented with natural concentrations of the major isothiocyanate formed by *A. thaliana* Col-0 leaves (*Figure 4*). These results support the involvement of this isothiocyanate in the negative consequences of *A. thaliana* aliphatic glucosinolates on silenced *P. xylostella*. Infiltration of ITCs into detached leaves only approximates the natural larval feeding conditions, and could for example lead to unnatural concentrations of ITC-derived products or alter larval feeding behavior. Nevertheless, the combined amounts of 4MSOB-ITC and its known conjugates in larvae after ingestion (*Figure 4—figure supplement 1*) very closely matched not only the natural concentrations of those compounds but also the total 4MSOB-ITC dose administered to leaves, suggesting that little additional metabolism had taken place. Additionally, the relative differences in larval weights between *gss*-silenced and control larvae feeding on wild-type *A. thaliana* leaves (approximately 40% lower at eight dph, *Figure 2A*) and between larvae fed on 4MSOB-ITC-infiltrated and non-infiltrated leaves (approximately 40–50% lower at eight dph, *Figure 4A*) are also similar, supporting an equivalence of the treatments.

At the time of its initial discovery in *P. xylostella* (*Ratzka et al., 2002*), desulfation of glucosinolates was recognized as a way for insect herbivores to avoid glucosinolate toxicity by preventing myrosinase catalysis of glucosinolate hydrolysis since desulfo-glucosinolates are known not to be substrates for myrosinases (*Matile, 1980*). The original report calculated that the GSS activity in each larva was sufficient to readily desulfate all ingested glucosinolates (*Ratzka et al., 2002*), and more recent quantitative isotopic tracer experiments revealed that over 80% of the glucosinolates ingested by *P. xylostella* are indeed desulfated (*Jeschke et al., 2017*). Here we show that desulfation in *P. xylostella* is a detoxification reaction that increases performance and reproductive fitness, in spite of the energy investment required. This conclusion is consistent with a recent evolutionary study (*Heidel-Fischer et al., 2019*) demonstrating that the *gss* genes of *P. xylostella* acquired their

present function encoding GSSs under positive selection pressure. Thus it seems likely that the glucosinolate desulfation reactions reported in the desert locust *S. gregaria* (*Falk and Gershenzon, 2007*), the turnip sawfly *Athalia rosae* (*Opitz et al., 2011*), the silverleaf whitefly *Bemisia tabaci* (*Malka et al., 2016*) and the flea beetle *Psylliodes chrysocephala* (*Beran et al., 2018*) also function as genuine detoxification measures.

Nevertheless, the desulfation process may be not equally effective for all types of glucosinolates. The 132 documented natural glucosinolates are classified as aliphatic, indolic and benzenic depending upon their side-chain structures, with further diversification from added double bonds, hydroxyl or carbonyl groups, and sulfur oxidation (*Agerbirk and Olsen, 2012*). A recent characterization of *P. xylostella* GSS enzymes indicates that the three individual GSS enzymes desulfate specific subsets of glucosinolate substrates, with GSS1 being particularly important for desulfation of aliphatic glucosinolates (*Heidel-Fischer et al., 2019*). Accordingly, the transcripts of *gss1* are higher than *gss2* and *gss3* in the midgut of *P. xylostella* larvae, consistent with aliphatic glucosinolates being the most abundant class in *A. thaliana* Col-0 plants (*Figure 1A* and *Figure 1—figure supplement 1*). Selective knock-out of *gss* genes may be a promising approach to more precisely determine the substrate range and efficiency of the GSS enzymes in vivo and provide more information about the relative toxicities of different glucosinolate types.

Herbivores feeding on glucosinolate-containing plants possess other mechanisms to avoid glucosinolate toxicity, which could complement GSS (*Jeschke et al., 2016a*; *Winde and Wittstock, 2011*). In this study, we found no evidence that *P. xylostella* naturally makes significant use of sequestration or formation of nitriles in addition to desulfation for processing glucosinolates. However, there are strong hints that *P. xylostella* larvae do possess further adaptations to avoid glucosinolate toxicity. After *gss* silencing, we could still detect substantial quantities of intact glucosinolates in the larval midgut and hemolymph (*Figure 3A*), demonstrating that myrosinase activity is not fully efficient in the *P. xylostella* gut. Myrosinase inhibition was also indicated in frass where intact glucosinolates were also found, and spiking with 4MSOB resulted in the formation of 10-fold higher concentrations of desulfo-4MSOB than 4MSOB-ITC (*Figure 3—figure supplement 1*). Although inhibition of myrosinase activity was proposed as a way to prevent glucosinolate intoxication (*Jeschke et al., 2016a*), until now there has been no concrete evidence to support it. GSS itself may inhibit myrosinase action not only by diversion of the substrate for hydrolysis, but by the inhibitory effect of the sulfate moiety released from the desulfation process on myrosinase activity (*Shikita et al., 1999*).

## Predatory lacewing larvae that metabolize the isothiocyanates in their *gss*-silenced larval prey incur a cost in reduced growth, but avoid long-term effects on fitness

Certain species of insect herbivores are well known to actively accumulate plant glucosinolates as defenses against their enemies (*Beran et al., 2014*; *Francis et al., 2001*; *Kazana et al., 2007*; *Winde and Wittstock, 2011*). Glucosinolate-sequestering insects typically contain high concentrations of these compounds, with *B. brassicae* wingless aphids and *Phyllotreta striolata* beetles both sequestering upwards of 30 µmol glucosinolates/g (*Kazana et al., 2007*; *Beran et al., 2014*), and the sawfly *Athalia rosae* hemolymph having 10–30 µmol glucosinolates/g hemolymph (*Müller et al., 2001*). Both *B. brassicae* and *P. striolata* not only sequester glucosinolates but also produce their own glucosinolate-activating myrosinases (*Pontoppidan et al., 2001*; *Jones et al., 2001*; *Beran et al., 2014*), and glucosinolate hydrolysis has strong negative effects on their predators (*Francis et al., 2001*; *Kos et al., 2011*).

Even insect herbivores that do not sequester glucosinolates may pose risks for predators and parasitoids because of the transient presence of these compounds or their hydrolysis products in their bodies. In this study, *gss*-silenced *P. xylostella* larvae fed on plants containing wild-type glucosinolate levels retained approximately 450 nmol/g fresh weight 4MSOB-ITC in larval midgut cells and ~300 nmol/mL in hemolymph (*Figure 3*), at least a 10-fold increase over non-silenced larvae. The predatory larvae of the common green lacewing *C. carnea* then detoxified these compounds via the general mercapturic acid pathway (*Figure 5A*), a process used by many generalist herbivores (*Schramm et al., 2012*) with the remaining isothiocyanates being deposited in the anal secretion (*Figure 5C*). However, *C. carnea* larval development was reduced on diets of *gss*-silenced *P. xylostella* (*Figure 6A*) likely reflecting the metabolic cost of detoxification and transport. In a previous study, the generalist herbivore *S. littoralis* was shown to suffer up to a 50% decline in growth rate on

ITC-containing diet, due to the diversion of cysteine towards biosynthesis of the glutathione required for isothiocyanate detoxification (*Figure 5A*), instead of towards protein synthesis (*Jeschke et al., 2016b*). However, *C. carnea* ingested lower levels of ITCs in its *P. xylostella* prey than those found in the guts of herbivores feeding on glucosinolate-containing plants, and so in spite of slower larval growth, no negative long-term effects on survival and fitness were observed (*Figure 6A–D*).

Besides functioning in the excretion of uric acid and production of prepupal silk (*Craig and Phillip, 1987*), the anal secretion of *C. carnea* also plays a role as a defense droplet which is transferred to the head or antennae of attacking enemies (*McEwen et al., 2001*). The presence of glucosinolate-derived isothiocyanates in the anal secretion might contribute to its defensive properties since isothiocyanates are repellent to several animals (*Newman et al., 1992*), but this topic needs further investigation. The detoxification ability of herbivore enemies like lacewings may also benefit the plant, since predatory insects can contribute to significant reductions in insect herbivore damage (*Price et al., 1980*). Interestingly, the isothiocyanate hydrolysis products of plant glucosinolates are reported to attract other herbivore enemies, parasitoids of insect herbivores that feed on glucosinolate-containing plants (*Blande et al., 2007*; *Gols and Harvey, 2009*).

## Conclusion and perspectives

The enzymatic desulfation of glucosinolates by *P. xylostella* has been long thought to allow these insects to suppress the activation of glucosinolate hydrolysis and thus circumvent the glucosinolate-myrosinase defense system of their host plants. By silencing the genes encoding GSSs in this insect, we demonstrated that desulfation significantly increases *P. xylostella* larval growth, survival and reproductive fitness. Given the metabolic costs of desulfation noted above and the possibility that this process is not the only mechanism present to avoid glucosinolate toxicity in *P. xylostella*, silencing or other genetic manipulation was necessary to unequivocally demonstrate its benefits to the organism. Our ability to effectively target the *gss* genes of this insect by plant-mediated RNAi suggests that application of RNAi via crop plants or by direct spraying (*Zhang et al., 2017*; *Zotti et al., 2018*), possibly in combination with increased glucosinolate levels in plant tissue, would significantly reduce *P. xylostella* damage by negatively impacting growth, survival and reproduction, as we observed in the laboratory. Additionally, we observed that a generalist larval predator was well-equipped to deal with the increased toxin content of its silenced prey. The lacewing efficiently detoxified and stored the ingested toxins with potential benefits for its own defense. It will be interesting to determine whether more specialized *P. xylostella* predators, which are not normally exposed to glucosinolate hydrolysis products, are also able to tolerate these plant-derived toxins.

## Materials and methods

### Plants

*Arabidopsis thaliana* Columbia-0 (Col-0) accession (with wild-type glucosinolates) and transgenic *myb28myb29* (without aliphatic glucosinolates) knockout mutant plants (*Sønderby et al., 2007*) were used. Plants were grown in a controlled short-day environment chamber at 21°C, 60% relative humidity, and a 14:10 hr light:dark photoperiod. *Brassica napus* for maintenance of insect cultures was grown in a controlled environment greenhouse under the same conditions.

### Insects

*Plutella xylostella* colonies, obtained from Bayer (Monheim am Rhein, Germany), were fed on *B. napus* leaves and maintained in a controlled short-day environment chamber. Eggs of *P. xylostella* were collected for experiments by inducing females to lay on a sheet of parafilm placed above *B. napus* leaves in colonies for two days. After hatching, *P. xylostella* larvae to be used in experiments were fed on *A. thaliana* plants growing under controlled short-day conditions, as described above. *Chrysoperla carnea* eggs were purchased from Katz Biotech AG (Baruth, Germany) and maintained in a controlled long-day environment chamber at 21°C, 70% relative humidity, and a 16:8 hr light:dark photoperiod.

### RNA isolation and cDNA synthesis

Midguts of *P. xylostella* fourth-instar larvae, which abundantly express *gss* (*Ratzka et al., 2002*), were dissected and pooled into TRIzol reagent (Invitrogen, Waltham, MA, USA), and then kept at 4° C before use. Total RNA was isolated from stored midguts according to the manufacturer's protocol and was subjected to DNaseI (Thermo Fisher Scientific, Waltham, MA, USA) treatment to eliminate genomic DNA contamination. cDNA was synthesized from this RNA by SuperScript III Reverse transcriptase kits (Invitrogen).

### *gss1* silencing construct for plant-mediated RNAi

The complete coding sequences of *P. xylostella gss* genes were retrieved from *Heidel-Fischer et al. (2019)*. A 526 bp stretch of *gss1* was selected and cloned from *gss1* in a synthesized cDNA pool obtained from primer pairs *gss1*F and *gss1*R (*Supplementary file 1*). *Xba*I and *Eco*RI restriction enzyme cutting sites were added to the ends of the selected *gss1* fragment using the primer pairs *gss1*VF and *gss1*VR (*Supplementary file 1*), and the fragment was further digested by *Xba*I and *Eco*RI (Thermo Fisher Scientific). The pTRV1 (YL192) and pTRV2 (YL156) vectors, which have been described previously (*Burch-Smith et al., 2006*), were used to establish a virus-based dsRNA producing system for plant mediated RNAi. The restriction-digested 526 bp *gss1* fragment was inserted into the *Xba*I-*Eco*RI-cut pTRV2 multiple cloning site by T4 DNA ligase (Invitrogen). pTRV2 containing the 526 bp *gss1* fragment was then used for the *gss1* RNAi construct and the empty vector without insert was used for construction of the empty vector control construct. Simultaneously, a pTRV2 (YL154) vector carrying a *phytoene desaturase* (*pds,* which leads to albino patches when silenced and can therefore serve as a positive control of transformation) insert, was prepared as a visible positive control of virus induced gene silencing.

### Plant transformation

pTRV1 and pTRV2 containing *gss1*, empty vector, and *pds* constructs were transformed into *Agrobacterium tumefaciens* strain GV3101. Cultures of *A. tumefaciens* were grown overnight at 28°C in LB medium containing 50 mg/L kanamycin and 50 mg/L gentamycin. The next day, *A. tumefaciens* cells were harvested by centrifugation (Avanti J-20XP, Beckman Coulter, Krefeld, Germany) at 4000 x g for 30 min. The collected cells were resuspended in infiltration medium (10 mM MgCl$_2$, 10 mM MES, and 150 µM acetosyringone in milli-Q water), adjusted to an OD$_{600}$ of 1.5 with a UV/Vis spectrophotometer (Ultrospec 2100 pro, Biochrom US, Holliston, MA, USA), and incubated at room temperature for 4 hr. *A. tumefaciens* harboring pTRV1 and pTRV2 containing *gss1*, empty vector, or *pds* constructs were mixed in equal amounts before infiltration in plants. Infiltration was performed with a needleless 1 mL syringe into three leaves of the four to five leaf stage (approximately 15 days post seed germination) of *A. thaliana* Col-0 and *myb28myb29* plants. Four weeks after infiltration, the albino patches caused by photobleaching due to reduced *pds* levels in *pds*-construct infiltrated plants were employed as a silencing marker.

### Plant metabolite extraction and HPLC analysis

Fifty-day old leaves of untreated, empty vector, and *gss1* RNAi infiltrated *A. thaliana* Col-0 and *myb28myb29* plants were collected in Falcon tubes and frozen in liquid nitrogen immediately. Leaves were freeze-dried (LPHA 1–4 LDplus freeze dryer, Martin Christ, Osterode am Harz, Germany) for 36 hr, and then homogenized by shaking with 3–4 metal balls (3 mm) in each tube. Glucosinolates, flavonoids, and phenylpropanoids were extracted from approximately 12 mg samples using 1 mL extraction solvent (80% methanol) with 50 µM sinalbin as an internal glucosinolate standard. After 5 min incubation on a horizontal shaker (230 rpm) with solvent, supernatants were collected by centrifugation (18,000 x g). Sequentially, 800 µL of each supernatant were loaded on DEAE-Sephadex A-25 columns (Sigma-Aldrich, Munich, Germany). The collected flow-through was diluted (1:3) with milliQ water for further flavonoid and phenylpropanoid measurements. The glucosinolates bound to DEAE-Sephadex, and were desulfated by 30 µL sulfatase treatment (for preparation of sulfatase solution see *Graser et al., 2000*) overnight at room temperature. The next day, desulfo-glucosinolates were eluted with 500 µL milliQ water. All collected fractions were stored at −20°C until analysis.

Desulfo-glucosinolates, flavonoids, and phenylpropanoids were analyzed on an Agilent Technologies 1100 Series HPLC (Agilent Technologies, Santa Clara, CA, USA) with a diode-array detector using a Nucleodur Sphinx RP column (250 × 4.6 mm×5 µm, Macherey-Nagel, Düren, Germany). Desulfo-glucosinolates were detected at 229 nm and quantified according to Burow et al. (2006). Water and acetonitrile were employed as mobile phases A and B, respectively. The elution profile was: 0–1 min, 1.5% B; 1–6 min, 5% B; 6–8 min, 7% B; 8–18 min, 21% B; 18–23 min, 29% B; and 23.1–24 min, 100% B; 24.1–28 min, 1.5% B, at a flow rate of 1.0 mL/min. Flavonoids and phenylpropanoids were detected at 330 nm according to the method described by Onkokesung et al. (2014) with 0.2% formic acid and acetonitrile employed as mobile phases A and B, respectively. The elution profile was: 0–20 min, 0% B; 20–20.1 min, 45% B; 20.1–22.1 min, 100% B; 22.1–26 min, 0% B, at a flow rate of 1.0 mL/min. All compounds were identified by comparison of retention times with those of authentic standards.

## Collection of *P. xylostella* tissues

Larvae of *P. xylostella* were continuously fed on untreated, empty vector, and *gss1* RNAi infiltrated *A. thaliana* Col-0 and *myb28myb29* plants in transparent boxes in a controlled environment chamber under conditions as described above. Each box contained one plant and up to 100 larvae, with fresh plants provided as necessary to ensure feeding ad libitum. Tissues from ten larvae were pooled to produce one sample. Midgut epithelium, hemolymph, integument and fat body, and frass were collected from fourth-instar larvae, totaling approximately 5 mg, 5 µL, 10 mg and 10 mg respectively (fresh weights). Hemolymph was collected by a 2 µL pipette through a small wound scratched by a 5 mm needle. Midguts were dissected in TE buffer (Tris- EDTA buffer, pH 8.0) under a dissecting microscope and the content of the midgut was carefully removed. Dissected midgut epithelium, integument and fat body were carefully washed in TE buffer to remove any adhering plant material and hemolymph. Pupae were collected on the second day after pupation and kept individually in 1.5 mL tubes. The pupal excretion (approximately 5 mg from five pupae combined) was collected simultaneously. Adult moths were collected immediately after emergence, divided by sex, and paired for mating (pairs of female and male moths from the same treatments). Subsequently, eggs were collected on the second day after mating (around 5 mg fresh weight per sample). All collected larval tissues, pupae and adults were weighed and immediately frozen in liquid nitrogen, and then stored at −80°C until further use.

## Quantitative real-time PCR (qPCR)

To conduct tissue-specific *gss* transcript profiling and quantify *gss* silencing efficiency, *gss* transcripts were quantified in *P. xylostella* fourth-instar larvae. RNA isolation and cDNA syntheses were performed as mentioned above. qPCR was performed to measure *gss* transcripts in the cDNAs, as reported (Senthil-Kumar and Mysore, 2014) by using qRT-PCR SYBR Green kit (Agilent Technologies). Meanwhile, transcripts of detoxification related genes (belonging to the P450, GST, CoE, UGT, sulfatase and SUMF families) were measured in the midgut epithelium of *P. xylostella* fourth-instar larvae by qPCR. The ubiquitin gene was used as an internal control to normalize the abundance of the other gene transcripts. All gene accession numbers and primer pairs were designed via Primer 3 software version 4.0 and listed in (Supplementary file 1).

## In situ hybridization

PCR products of *P. xylostella gss1*, *gss2* and *gss3* coding sequences were sequenced and then cloned into pGEM-T vectors (Promega, Wisconsin, USA), which were subsequently subjected to in vitro transcription. The linearized pGEM-T vectors consisting of *P. xylostella gss* coding sequences were utilized to synthesize both sense and antisense riboprobes labeled with digoxigenin (Dig) or biotin (Bio) using the T7/SP6 RNA transcription system (Roche, Mannheim, Germany).

Fourth-instar *P. xylostella* larvae were isolated and starved for 1 hr prior to the tissue preparation. Freshly dissected *P. xylostella* midguts were embedded in Tissue-Tek O.C.T. compound (Sakura Finetek Europe, The Netherlands). Transverse cryosections with the thickness of 16 µm were thaw mounted on SuperFrost Plus slides (Menzel-Gläser, Braunschweig, Germany) at −21°C (Jung CM300 cryostat). RNA In situ hybridization was performed as previously reported (Jiang et al., 2018) with slight modifications. In brief, the cryosections were first fixed (4% paraformaldehyde in 0.1 M

NaHCO₃, pH 9.5) at 4°C for 25 min, then were subjected to a series of treatments at room temperature: a wash for 1 min in PBS (phosphate buffered saline: 0.85% NaCl, 1.4 mM KH$_2$PO$_4$, 8 mM Na$_2$HPO$_4$, pH 7.1), an incubation for 10 min in 0.2 M HCl, another two washes for 1 min in PBS, an incubation for 10 min in acetylation solution (0.25% acetic anhydride freshly added to 0.1 M triethanolamine) and three washes in PBS (3 min each). Afterwards, the sections were pre-hybridized for 15 min at 4°C bathed in hybridization buffer (50% formamide, 5x SSC, 50 µg/mL heparin, and 0.1% Tween-20). A volume of 100 µL hybridization solution containing assayed sense or antisense riboprobes in hybridization buffer was evenly applied onto the tissue section. A coverslip was placed on top and slides were incubated in a moisture box at 60°C overnight (18–20 hr). On the second day, slides were washed twice for 30 min in 0.1x SSC at 60°C. Then each slide was treated with 1 mL 1% blocking reagent (Roche) for 40 min at room temperature.

Visualization of hybridized riboprobes was achieved by using an anti-Dig AP-conjugated antibody in combination with HNPP/Fast Red (Roche) for Dig-labeled probes and an anti-biotin streptavidin horse radish peroxidase-conjugate together with fluorescein-tyramides as substrate (TSA kit, Perkin Elmer, MA, USA) for biotin-labeled probes. Cytox green was diluted to 1:30000 in PBS buffer for nuclei counter staining, and each slide was covered with a 100 µL mixture of these components and incubated for 3 min at room temperature. Fluorescence signals were analyzed with a Zeiss LSM510 Meta laser scanning microscope (Zeiss, Oberkochen, Germany), and the acquired confocal image stacks were processed by ZEN 2009 software. The images presented in this paper were rendered by a projection of several optical planes selected from a range of confocal image stacks. For clearer presentation, images were slightly adjusted in brightness and contrast.

### In vitro enzyme assay

Midguts of *P. xylostella* larvae feeding on empty vector and *gss1* RNAi infiltrated Col-0 and *myb28-myb29* plants were homogenized in Tris buffer (100 mM, pH 7.5). Concentrations of protein were measured using the BCA Protein Assay Macro Kit (Serva Electrophoresis, Heidelberg, Germany). Protein (1 µg amounts) from each sample in 50 µL Tris buffer was incubated with 50 µL 10 mM 4MSOB (Carl Roth, Karlsruhe, Germany) for 3 min at 25°C. The reaction was stopped by adding 500 µL methanol. Subsequently, the concentration of desulfo-4MSOB formed was determined by LC-MS/MS (Agilent 1200 series-API3200). In vitro enzyme activity of I3M (Carl Roth), used as a representative of indolic glucosinolates, was measured in the same way.

### Growth, development and reproduction of *P. xylostella*

To determine the impact of downregulated *gss* transcripts on *P. xylostella* performance, we monitored the weight of larvae, the mortality of pupae, and the egg-laying capacity and egg-hatching percentage of adults raised from treated larvae fed continuously on empty vector and *gss1* RNAi infiltrated Col-0 and *myb28myb29* plants prepared as described above. Firstly, weights of treated larvae were measured at 4, 6 and 8 dph (days post hatching). After pupation, around 50 pupae from each treatment were collected to monitor pupal mortality. The experiment was repeated independently three times (see *Figure 2—source data 1*). To inspect the reproduction of *P. xylostella*, male and female fourth-instar larvae were separated (based on the light yellow spot caused by the testicle on the fifth abdominal segment of male larvae), and resulting pupae were paired in individual 35 mL plastic vials. Pupae usually emerged on the same day. If not, according to the their genitals, the newly emerged moth was paired with another newly emerged moth (if one was available) in a new vial, or discarded. Specifically, female moths from each treatment were separated and paired with male moths from larvae feeding on empty vector infiltrated Col-0 plants; in parallel male moths from each treatment were paired with female moths from larvae feeding on empty vector infiltrated Col-0 plants (*Supplementary file 2*). Paired moths were kept in 35 mL plastic vials with 6% sugar solution. Thirty replicates (pairings) were performed for each treatment, and successful mating in the first day post emergence was recorded. Two days post mating, the numbers of eggs laid by 20 of the successfully mated couples from each treatment were counted. The hatching percentage of those eggs was recorded 4 days later. As an additional control, treated moths mated with moths from larvae feeding on untreated Col-0 or *myb28myb29* plants were also studied (*Supplementary file 2*).

## Metabolite extraction and LC-MS/MS analysis

Samples of *P. xylostella* tissue for targeted analyses of metabolites collected as described above (subsection 'Collection of *P. xylostella* tissues') were homogenized in 200 µL extraction solvent (60% methanol, pH 3.0) in 1.5 mL Eppendorf tubes with ceramic beads (Sigmund Lindner, Warmenstei-nach, Germany) by a Skandex S-7 homogenizer (Grootec GmbH, Kirchheim, Germany) for 3 min. Homogenized samples were centrifuged at 13,000 x g for 20 min at room temperature to separate undissolved particles. Clear supernatants were transferred to 2 mL amber glass vials with 0.3 mL glass inserts and further analysed by LC-MS/MS to determine glucosinolate, desulfo-glucosinolate and isothiocyanate concentrations. Analyses were performed on an Agilent Technologies 1200 Series HPLC (Agilent Technologies) coupled to an API 3200 triple-quadrupole mass spectrometer (Applied Biosystems Sciex, Darmstadt, Germany). 4MSOB and desulfo-4MSOB were analyzed by loading samples onto a Nucleodur Sphinx RP column (250 × 4.6 mm×5 µm, Macherey-Nagel, Düren, Germany) with mobile phase A (0.2% formic acid in milliQ water) and mobile phase B (acetonitrile). The elution profile was: 0–1 min, 1.5% B; 1–6 min, 5% B; 6–8 min, 7% B; 8–9 min, 8.4% B; 9.1–10 min, 100% B; and 10.1–14 min, 1.5% B, at a flow rate of 1.0 mL/min. 4MSOB–ITC and its conjugates were analyzed by loading samples onto a Agilent Zorbax Eclipse XDB-C18 column (50 × 4.6 mm×1.8 µm, Agilent Technologies, Wilmington, DE, USA) with mobile phase A (0.05% formic acid in milliQ water) and mobile phase B (acetonitrile). The elution profile was: 0–0.5 min, 15% B; 0.5–2.5 min, 85% B; 2.5–3.5 min, 100% B; 3.5–6.0 min, 3.0% B, at a flow rate of 1.0 mL/min. I3M and desulfo-I3M were analyzed by loading samples onto the same column with the same mobile phases with an elution profile of: 0–0.5 min, 5% B; 0.5–4.0 min, 60% B; 4.1–6.0 min, 100% B; 6.1–8.5 min, 5% B, at a flow rate of 1.0 mL/min. I3C and its derivatives were analyzed by loading samples onto the same column with mobile phase A (10 mM ammonium formate in milliQ water) and mobile phase B (acetonitrile). The elution profile was: 0–0.5 min, 10% B; 0.5–6 min, 90% B; 6.1–7.5 min, 100% B; 7.6–10.0 min, 10% B, at a flow rate of 1.0 mL/min. Quantification of each compound was achieved by multiple reaction monitoring (MRM) of specific parent to product ion conversions for each compound. Parameters for 4MSOB, desulfo-4MSOB (*Malka et al., 2016*), 4MSOB–ITC and its conjugates (*Gloss et al., 2014*); and for I3M and desulfo-I3M (*Malka et al., 2016*) were previously determined, while those for I3C and its derivatives are given in *Supplementary file 3*. Analyst 1.5 software (Applied Biosystems) was used for data acquisition and processing. Quantification of individual compounds was achieved by external calibration curves, the origin of the external standards are given in *Supplementary file 4*.

## Frass spiking assay

Larvae of *P. xylostella* were continuously fed on *myb28myb29* mutant plants (without aliphatic gluco-sinolates) in transparent boxes (*n* = 6 boxes), each box containing one plant and around 50 larvae. Fresh plants were provided every day until larvae reached the fourth-instar stage. Then, fresh larval frass was collected (one separate sample per box) and immediately frozen in liquid nitrogen for fur-ther use. Two frass aliquots (approximately 0.5 mg each) were weighed from each frass stock for enzyme assays. One aliquot was heated at 100°C for 30 min as a negative control. Both heat-inacti-vated and fresh frass sets were spiked with 4MSOB (10 µL of 2.5 mM) and incubated for 15 min at room temperature. The reaction was stopped by adding 100 µL methanol. Subsequently, concentra-tions of 4MSOB, desulfo-4MSOB and 4MSOB-ITC were measured by LC-MS/MS to determine *P. xylostella* GSS and plant myrosinase activities.

## Complementation experiments

To determine whether formation of 4MSOB-ITC in *P. xylostella* caused the phenotypes attributed to *gss* silencing, complementation experiments were conducted by infiltrating 4MSOB-ITC (BIOZOL Diagnostica Vertrieb, Eching, Germany) into empty vector and *gss1* RNAi infiltrated *myb28myb29* leaves. Natural concentrations of 4MSOB-ITC were employed as would result from hydrolysis of typi-cal *A. thaliana* Col-0 foliage: 250 nmol/g fresh weight leaf, as determined by LC-MS/MS (*Figure 4—figure supplement 1*). Thus 0.3 µL of 800 µM 4MSOB-ITC per mg fresh leaf was infiltrated in a sol-vent of 0.4% aqueous ethanol. Leaves infiltrated with solvent alone served as negative controls (*Fig-ure 4—figure supplement 1*). Larvae of *P. xylostella* were continuously fed on these leaves from

hatching. Larval weight, pupal mortality, egg-laying capacity and egg-hatching percentage were recorded as previously mentioned.

## Collection of *C. carnea* tissues

Larvae of *C. carnea* were fed continuously with *P. xylostella* larvae that in turn fed on empty vector or *gss1* RNAi infiltrated Col-0 or *myb28myb29* plants. Each *C. carnea* larva was separately fed on *P. xylostella* larvae from the corresponding groups in a 35 mL vial. Numbers and developmental stages of the prey were chosen according to the predator developmental stage to ensure *C. carnea* always had sufficient food. First- and second-instar *C. carnea* larvae were fed on second- and fourth-instar *P. xylostella* larvae, respectively. When *C. carnea* larvae reached third-instar, guts with gut lumen, hemolymph, Malpighian tubules, anal secretion, and integument and fat bodies were collected, totaling approximately 3 mg, 1 µL, 0.5 mg, 3 µL and 5 mg respectively (fresh weights). Tissues of three larvae were pooled to produce one sample. The anal secretion of *C. carnea* larvae was collected with a 10 µL pipette. Larvae transfer the droplet to the pipette tip as a defense reaction when touched by the tip on the dorsal abdomen. Collected anal secretion was washed into 200 µL extraction solvent (60% methanol, pH 3.0) immediately and kept at −20°C until further analyses. Hemolymph of each larva was collected by a 10 µL pipette through a small wound in the posterior abdominal segment scratched by a 5 mm needle. Tissues were dissected under a dissecting microscope and washed in TE buffer carefully to remove adhered hemolymph. Pupal pellet left in the cocoon was collected after adult emergence (approximately 1 mg from three pupae combined). Meconium (approximately 5 mg from three adults combined) excreted by the adults in the first few hours after emergence was collected together with the adults. All collected larval, pupal and adult tissues and excretions were weighed and immediately frozen in liquid nitrogen, and then stored at −80°C until further analyses.

## Performance of *C. carnea*

To determine the influence of 4MSOB-ITC on *C. carnea*, the performance of this lacewing was assessed when continuously preying on *P. xylostella* fed on empty vector or *gss1* RNAi infiltrated plants from either Col-0 or *myb28myb29* backgrounds. Newly hatched *C. carnea* (0 dph, days post hatching) were fed with second-instar *P. xylostella* larvae from each treatment in individual 35 mL plastic vials ($n \geq 60$ per treatment). To examine *C. carnea* larval growth (*Figure 6A*) and pupation (*Figure 6B*), a subset of *C. carnea* larvae ('development subset', $n = 30$ in individual vials for each treatment) was separated five dph and observed until pupation while continuously feeding on fourth-instar *P. xylostella* larvae from each treatment. *C. carnea* larval weights were recorded 5, 8, 11 and 13 dph, and the number of pupae was recorded from 13 dph (day when the first larva pupated) until 24 dph (last larva pupated). The *C. carnea* larvae not included in the development subset were also fed continuously on *P. xylostella* larvae from the respective treatments, and surviving insects from the two *C. carnea* subsets were combined per treatment upon pupation to record pupal mortality (*Figure 6C*, $n = 30$–33). Pupal mortality measurements were repeated two additional times with independent groups of pupae (see *Figure 6—source data 1*) fed as above, but without recording of larval development. Numbers of eggs laid (*Figure 6D*) were recorded from an independent batch of insects raised for this purpose alone: freshly emerged adults were sexed, and couples (females and males originating from the same treatment and hatched on the same day) were paired in 500 mL plastic boxes with 6% honey solution ($n = 14$–17 couples per treatment). The number of eggs laid was counted on the fourth day post mating.

## Predation bioassay for *C. carnea*

Choice and no-choice assays were conducted in 35 mL plastic vials. Third-instar larvae of the lacewing *C. carnea* were used that had been feeding on *P. xylostella* larvae fed in turn on empty vector infiltrated Col-0 plants. Fourth-instar larvae of *P. xylostella* fed on empty vector or *gss1* RNAi infiltrated plants from either Col-0 or *myb28myb29* backgrounds were used as prey. Lacewing larvae were starved for 12 hr before the assay and not reused after testing. In choice assays (*Figure 6E*, 27-44 replicates for each treatment, with three independent experiment repetitions, see *Figure 6— source data 1*), each *C. carnea* larva was allowed to choose between two test *P. xylostella* larvae from different treatments. Each assay consisted of 20 min of observation, and data were not

collected from *C. carnea* that did not make a choice after 20 min; that is, only data from *C. carnea* larvae that captured and killed a *P. xylostella* larva within 20 min were recorded. In no-choice assays (*Figure 6F*, 21–23 replicates for each treatment, with four independent experiment repetitions, seen in *Figure 6—source data 1*), one *P. xylostella* larva was offered to a *C. carnea* larva in each assay container, and the *P. xylostella* larvae captured and killed within 20 min were counted. In a larger scale bioassay (*Figure 6G*), the percentage of *P. xylostella* larvae on a plant that were preyed upon by the lacewing *C. carnea* was determined. *P. xylostella* fourth-instar larvae (40 larvae for each treatment, with two independent experiment repetitions, see *Figure 6—source data 1*) fed since hatching on empty vector- or *gss1* RNAi-infiltrated plants from either Col-0 or *myb28myb29* backgrounds were transferred to intact *A. thaliana* plants of the corresponding genotypes a few hours before the assay. Five third-instar *C. carnea* larvae were then placed on each plant, and the numbers of remaining *P. xylostella* larvae were counted after 24 hr.

## Statistical analyses

Data were analyzed using the SPSS statistics package version 17.0 and R version 3.6.1. Figures were created using Origin 2019. All data were checked for statistical prerequisites such as homogeneity of variances and normality. Quantitative data (gene transcripts in the larval midgut, in vitro enzyme assays, metabolites in leaf and larval tissues, *C. carnea* larval weights, and egg numbers) were analyzed by one-way ANOVA; multiple comparisons (*gss* gene transcripts in different larval tissues, enzyme activities in the frass spiking assay, and *P. xylostella* larval weights) were analyzed by two-way ANOVA; and statistically significant differences ($p \leq 0.05$) between means (±s.e.) were determined by Tukey HSD tests. Significance ($p \leq 0.05$) of the binary results of mortality and predation assays was evaluated using a two-sided proportions test. *P. xylostella* egg hatching data were analyzed by Dunn's post hoc tests in conjunction with non-parametric Kruskal-Wallis tests. Significance ($p \leq 0.05$) of *C. carnea* larval pupation was determined by Kaplan-Meier survival analysis. *C. carnea* choice assays between caterpillar sources were analyzed with a two-sided binomial test, and effects between each treatment were analyzed by a generalized linear model (GLM) with a binomial distribution and a logit link function. Statistical tests and numbers of replicates are provided in the figure legends and Figure – Source Data files. Letters in graphs represent $p \leq 0.05$; asterisks represent *$p \leq 0.05$, **$p \leq 0.01$, ***$p \leq 0.001$; and data groups in each panel that are not labeled with asterisks or letters are not statistically different from each other.

## Data and materials availability

All the data needed to understand and assess the conclusions of this research are available in the manuscript; additional data and materials related to this paper may be requested from the authors.

## Acknowledgements

We thank Dr. Heiko Vogel and Dr. Roy Kirsch for generously providing *gss* sequences and helpful discussions, Bettina Raguschke for technical assistance, Maria K Paulmann for statistics advice, Dr. Sascha Eilmus from the Bayer Corporation for supplying *Plutella xylostella*, and the Max Planck Society Partner Group Fund for support of Sagar Pandit.

## Additional information

### Funding

| Funder | Author |
| --- | --- |
| China Scholarship Council | Ruo Sun |
| Max-Planck-Gesellschaft | Ruo Sun<br>Xingcong Jiang<br>Michael Reichelt<br>Jonathan Gershenzon<br>Sagar Subhash Pandit<br>Daniel Giddings Vassão |

The funders had no role in study design, data collection and analysis, decision to publish, or preparation of the manuscript.

## Author contributions
Ruo Sun, Conceptualization, Resources, Data curation, Formal analysis, Validation, Investigation, Visualization, Methodology, Project administration; Xingcong Jiang, Conceptualization, Resources, Data curation, Formal analysis, Investigation, Visualization, Methodology; Michael Reichelt, Resources, Formal analysis, Validation, Investigation, Methodology; Jonathan Gershenzon, Conceptualization, Resources, Data curation, Supervision, Funding acquisition, Methodology, Project administration; Sagar Subhash Pandit, Conceptualization, Resources, Data curation, Supervision, Validation, Methodology, Project administration; Daniel Giddings Vassão, Conceptualization, Resources, Data curation, Supervision, Visualization, Methodology, Project administration

## Author ORCIDs
Ruo Sun https://orcid.org/0000-0001-9861-6097
Michael Reichelt http://orcid.org/0000-0002-6691-6500
Jonathan Gershenzon https://orcid.org/0000-0002-1812-1551
Daniel Giddings Vassão https://orcid.org/0000-0001-8455-9298

## Decision letter and Author response
Decision letter https://doi.org/10.7554/eLife.51029.sa1
Author response https://doi.org/10.7554/eLife.51029.sa2

# Additional files
## Supplementary files
• Supplementary file 1. Primer sets for gene cloning and qRT-PCR validation.

• Supplementary file 2. The sources of *Plutella xylostella* moths for adult fecundity experiments.

• Supplementary file 3. LC-MS/MS parameters used for the multiple reaction monitoring (MRM) analyses of I3C derivatives. Q1, quadrupole one voltage; Q3, quadrupole three voltage; DP, declustering potential; EP, entrance potential; CEP, collision cell entrance potential; CE, collision energy; CXP, collision cell exit potential.

• Supplementary file 4. External standards used for quantification.

• Transparent reporting form

## Data availability
All data generated or analysed during this study are included in the manuscript and supporting files. Source data files have been provided for figures and figure supplements.

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
