## [Decision Letter]

**Acceptance summary:**

Tritrophic interactions have a long history in studies of ecology and evolution, but the specific mechanisms-and much less the individual genes-driving these interactions are not well known. One particular candidate for mediating such interactions is a sulfatase gene in a plant herbivore that is believed to provide resistance to plant defenses and possibly impact predators. In this study, the authors utilize cross-species RNA approaches to abolish the sulfatase gene in the herbivore and showed that this alters the herbivores interaction with both the plant and the predator. This provides an avenue to begin mechanistic studies in tritrophic interactions where the system is recalcitrant to transformation or to other genetic interventions.

**Decision letter after peer review:**

Thank you for submitting your article "Multitrophic metabolism of plant chemical defenses and its effects on herbivore and predator performance" for consideration by *eLife*. Your article has been reviewed by three peer reviewers, one of whom is a member of our Board of Reviewing Editors, and the evaluation has been overseen by Detlef Weigel as the Senior Editor. The reviewers have opted to remain anonymous.

The reviewers have discussed the reviews with one another and the Reviewing Editor has drafted this decision to help you prepare a revised submission.

All the reviewers and editors agreed that the work is a very interesting and explicit test of how individual genes can influence multi-trophic interactions. Below are merely some specific suggestions for clarifications and more appropriate statistics that will help to strengthen the manuscript. They do not require any new experimentation. Please see the specific suggestions in the individual reviews below.

Reviewer #1:

In this manuscript, the authors utilize a trans-specific RNA to knockout a key defense mechanism within *P. xylostella* and then investigate how it affects multi-trophic interactions. This provides an interesting mechanistic assessment of multitrophic interactions via a specific enzyme/chemical relationship.

The authors should probably caveat the idea that the isothiocyanates are generally toxic and how broadly the work can be interpreted. For example, if this work was done in a plant accumulating the 2-hydroxy-but-3-enyl glucosinolate, the main hydrolysis product would actually be goitrin which has the ability to directly influence pupal transitions due to interacting with the juvenile hormone receptor that is a paralogue of the human thyroid receptor targeted by goitrin. Thus, I would encourage the authors to be more explicit about how shifting to different plants with different chemotypes might yield a different answer. Relying on the general isothiocyanates are generic toxins might be an over assumption in this system.

Infiltrating Arabidopsis with 4MSOB-ITC could be creating new compounds as Arabidopsis has processes to metabolize these compounds. For example, work by the Jeschke and Bednarek groups has shown that Arabidopsis will create raphanusamic acid from a wide range of ITCs. Thus, it is not clear how to truly interpret the subsection “4MSOB-ITC is responsible for the negative effects of gss silencing on *P. xylostella* performance”. This could be due to the 4MSOB-ITC or it could be from activities associated with the further catabolism of this product. While it appears as a nice complementation experiment, the authors should discuss the complexities of its interpretation.

Reviewer #2:

In this manuscript, Sun et al. performed VIGS transient knockdown of the *A. thaliana* plants, fed different transient mutants to *Plutella xylostella* larvae, compared glucosinolate metabolism in larval body across different genotypes, and probed for tritrophic interactions mediated by glucosinolates with *C. carnea* – a predator of *P. xylostella*. Authors found that *P. xylostella* glucosinolate sulfatase significantly reduces the toxic effects of isothiocyanates. Authors also suggest that *C. carnea* is able to withstand the toxic effects of glucosinolate hydrolysis by derivatizing isothiocyanates differently.

The manuscript is very well-written and experiments beautifully designed and executed. I have no major concerns about the manuscript. Some suggestions noted below:

– Unlike Figure 2C, D, F, distributions in Figure 2B are bimodal. Is that a statistical fluke? Any thoughts on why that is the case? If that experiment has not been replicated, I would suggest doing that to ensure consistency of results and robustness of the bimodal pattern, which could point to something interesting. This is not an essential experiment for the revision though.

– In Figure 5B, I'm assuming most Desulfo-4MSOB is from the ingested *P. xylostella* larvae? If so, that is not clear in the text. Also, given the levels of 4MSOB-ITC-NAC in Figure 5D in anal secretion are only 33-50% of the 4MSOB-ITC levels shown in Figure 5C, does it not mean that *C. carnea* manages 4MSOB-ITC mostly through excretion rather than detoxification? Figure 5—figure supplement 1 also suggests that only ~40% of the 4MSOB-ITC is detoxified to NAC, while the rest is excreted as such and minimally accumulates in the haemolymph.

– Roughly speaking, what% of the effects (none/some/most) of feeding 4MSOB-ITC to larvae (Figure 4) could be due to lower feeding frequency/changes in feeding behavior e.g. due to the feed becoming distasteful? Any previous experiments or evidence from this paper to argue against it? If not, please note the possibility in the manuscript.

Reviewer #3:

I was excited to read this interesting study examining the physiological and ecological consequences of detoxification of plant defenses in herbivores and their predators. The main finding is that gss-silencing reduced larval development and reproduction of a specialist herbivore. I was particularly interested in the effects of plant defenses cascading up to higher trophic levels. Lacewings fed on both silenced and non-silenced prey, and although there was a slight reduction in larval performance, gss-silencing had no effect on survival or reproduction. The question is novel with important implication to crop protection.

Overall the paper reads well, however some statistical analyses need clarification. A different statistical approach is recommended to test percentage data. To test differences in choice between different treatment the authors should use a GLM with binomial distribution and logit link function. Also, GLM with logit distribution would be a more robust test for percentages. Why was mortality/survival not analyzed using survival analysis?

In some experiments, it was unclear whether caterpillars were being fed individually or in groups. It is important to know how many plants and how many caterpillars were used because caterpillar feeding on the same plant are not independent and should not be treated as such in the analyses.

[Editors' note: further revisions were requested prior to acceptance, as described below.]

Thank you for submitting your article "Tritrophic metabolism of plant chemical defenses and its effects on herbivore and predator performance" for consideration by *eLife*. Your article has been reviewed by three peer reviewers, one of whom is a member of our Board of Reviewing Editors, and the evaluation has been overseen by Detlef Weigel as the Senior Editor. The reviewers have opted to remain anonymous.

The reviewers have discussed the reviews with one another and the Reviewing Editor has drafted this decision to help you prepare a revised submission.

Summary:

This work creates trans-kingdom RNAi to investigate the role of insect enzymes in detoxifying plant compounds. This provides some of the first gene level evidence for the mechanisms controlling multi-trophic interactions.

Essential revisions:

All the reviewers were happy with the revision but in discussions we all agreed that we should go with a revision to provide a chance to clarify some aspects surrounding the multi-modality. We agree that the biomodality is balanced across the groups so not changing the mean relationships. However, nearly ever reviewer and editor has been puzzled by this indicating that every reader will be equally puzzled. Any explicit thoughts that you can provide to explain this biomodality will go a long way to alleviating the reviewers puzzlement. Different mother ages? etc? Further, it would be advised to state that the residuals are normally distributed between the groups in these tests to alleviate statistical concerns.

Reviewer #1:

I would like to thank the reviewers for shifting to two-way ANOVAs but it is not clear how these are being presented in the figures. A two-way ANOVA will provide a significance for the main effects and the interaction while in the figures it still appears to be pairwise t-test results that are being shown.

I am still concerned by the bimodality in the egg laying/hatching experiment and the correlation analysis further raises this as a concern. In the correlation analysis, it appears that there are really two correlations in the data with one being a low hatching. Is there variation in the age of the mothers such that some are either much younger or older than others that could be linked to this?

Reviewer #2:

The authors have satisfactorily addressed all my concerns.

Reviewer #3:

The authors did a great job addressing all my comments.

---

## [Author Response]

Reviewer #1:[…] The authors should probably caveat the idea that the isothiocyanates are generally toxic and how broadly the work can be interpreted. For example, if this work was done in a plant accumulating the 2-hydroxy-but-3-enyl glucosinolate, the main hydrolysis product would actually be goitrin which has the ability to directly influence pupal transitions due to interacting with the juvenile hormone receptor that is a paralogue of the human thyroid receptor targeted by goitrin. Thus, I would encourage the authors to be more explicit about how shifting to different plants with different chemotypes might yield a different answer. Relying on the general isothiocyanates are generic toxins might be an over assumption in this system.

Thanks for pointing this out, we agree that our previous generalization could be wrong in some cases. We now try to be clearer (Introduction, first paragraph) that this general toxicity can be further modulated by the side-chain – and in cases such as the more reactive progoitrin and indolic glucosinolate products, would even result in products lacking a chemically exposed –N=C=S group. Although we focus here strongly on 4MSOB, these side-chain dependencies would be an interesting avenue to follow in more detail in the future.

Infiltrating Arabidopsis with 4MSOB-ITC could be creating new compounds as Arabidopsis has processes to metabolize these compounds. For example, work by the Jeschke and Bednarek groups has shown that Arabidopsis will create raphanusamic acid from a wide range of ITCs. Thus, it is not clear how to truly interpret the subsection “4MSOB-ITC is responsible for the negative effects of gss silencing on P. xylostella performance”. This could be due to the 4MSOB-ITC or it could be from activities associated with the further catabolism of this product. While it appears as a nice complementation experiment, the authors should discuss the complexities of its interpretation.

This is a good point and we have added a few sentences (subsection “Plant-mediated RNAi efficiently silences *P. xylostella gss* with severe physiological and fitness consequences”, second paragraph) on that regard. While ITC catabolites might be formed as a result of our infiltration, we expect their amounts and effects were small, as the amounts of ITC added to the leaves closely matched the amounts we recovered as the free ITC and known GSH-derived products. Unfortunately, our previous attempts at other delivery methods (e.g. coating the surface of leaves with ITC solutions) were not successful (leading to very non-uniform ITC distributions), so infiltration was the best available choice we had for a clean complementation experiment.

Reviewer #2:[…] The manuscript is very well-written and experiments beautifully designed and executed. I have no major concerns about the manuscript. Some suggestions noted below:– Unlike Figure 2C, D, F, distributions in Figure 2B are bimodal. Is that a statistical fluke? Any thoughts on why that is the case? If that experiment has not been replicated, I would suggest doing that to ensure consistency of results and robustness of the bimodal pattern, which could point to something interesting. This is not an essential experiment for the revision though.

The reviewer is likely referring to Figure 2E instead of Figure 2B, as the other reviewers did. As stated in the reply to reviewer #1, we cannot explain this distribution. This experiment was performed as a single batch and not subdivided into blocks. Although the origin of this distribution is currently unknown, this effect seems to be consistent, as similar distributions were observed when the experiment was performed with infiltrated ITC (Figure 4) and with *myb28myb29* plants (Figure 2—figure supplement 2) instead of Col-0 EV controls. As an attempt to present the data more clearly, we are now also showing the numbers of eggs hatched as a correlation to the numbers of eggs laid (Figure 2G, H; Figure 4G, H; and Figure 2—figure supplement 2I-L) – this way the lower hatching success can also be visualized as a lower slope of the corresponding line, and it is apparent that low-hatching batches were not clustered in particular egg numbers but rather distributed along the x axis. Regarding Figure 2B, measurements of pupal mortality were taken in 3 independent experiment repetitions (see Figure 2—source data 1) with consistent results.

– In Figure 5B, I'm assuming most Desulfo-4MSOB is from the ingested P. xylostella larvae? If so, that is not clear in the text. Also, given the levels of 4MSOB-ITC-NAC in Figure 5D in anal secretion are only 33-50% of the 4MSOB-ITC levels shown in Figure 5C, does it not mean that C. carnea manages 4MSOB-ITC mostly through excretion rather than detoxification? Figure 5—figure supplement 1 also suggests that only ~40% of the 4MSOB-ITC is detoxified to NAC, while the rest is excreted as such and minimally accumulates in the haemolymph.

Yes, this is our interpretation as well, that both desulfo-4MSOB and 4MSOB-ITC present in *C. carnea* were ingested from *P. xylostella* larvae. Thanks for pointing out that this was not clear in the text, we now added this information to the subsection “Lacewing larvae predating upon *gss*-silenced *P. xylostella* detoxify and mobilize activated glucosinolate hydrolysis products”.

The reviewer is correct on the second point – 4MSOB-ITC is more abundant in the anal secretion than its detoxification product 4MSOB-NAC, suggesting that much of the ingested ITC is efficiently transported/excreted. While we do not know the metabolic cost of ITC transport, we have studied the costs of 4MSOB-ITC metabolism (in terms of growth) via the mercapturic acid pathway (Figure 5A) in generalist herbivores, and found that the loss of GSH/Cys contributes greatly to the growth decrease (Jeschkeet al., 2016). Although the doses of 4MSOB-ITC ingested by herbivores and by *C. carnea* feeding on *gss*-silenced *P. xylostella* are different, the fraction of ingested ITC metabolized by *C. carnea* is larger than by the herbivores. In our interpretation, we propose that this metabolic route is also a contributing factor to the reduction in predator growth, and have now briefly mentioned that transport costs might also play a role.

– Roughly speaking, what% of the effects (none/some/most) of feeding 4MSOB-ITC to larvae (Figure 4) could be due to lower feeding frequency/changes in feeding behavior e.g. due to the feed becoming distasteful? Any previous experiments or evidence from this paper to argue against it? If not, please note the possibility in the manuscript.

While ITCs are attractive to ovipositing *P. xylostella* moths (Renwicket al., 2006), but lethal to larvae at higher concentrations (Liet al., 2000), to the best of our knowledge there are no reports directly examining ITC effects on *P. xylostella* larval feeding preference. Previous research on the roles of the glucosinolate-activating enzymes (myrosinases) on larval preference offers indirect but ambiguous evidence about the influence of ITCs. *P. xylostella* larvae fed equally well on wild-type *A. thaliana* and the *tgg1/tgg2* myrosinase knockout mutant (Badenes-Perezet al., 2013), suggesting that the naturally low amounts of glucosinolate activation and ITC formation do not play a role in preference. On the other hand, *B. juncea* lines with higher myrosinase activities supported lower insect feeding, but surprisingly had no effects on growth rates (Liet al., 2000).

In our studies, larvae appeared to feed continuously on leaves of all the treatments used throughout the experiment, but this was not measured. The decreases in growth of ITC-fed larvae relative to control-fed larvae were similar in magnitude to the decreases observed when feeding on wild-type versus *myb28myb29 A. thaliana*. Nevertheless, it’s indeed possible that some of the growth effects stem from lower preference/ingestion. We added a brief discussion of that point to the second paragraph of the subsection “Plant-mediated RNAi efficiently silences *P. xylostella gss* with severe physiological and fitness consequences” to reflect that.

Reviewer #3:[…] Overall the paper reads well, however some statistical analyses need clarification. A different statistical approach is recommended to test percentage data. To test differences in choice between different treatment the authors should use a GLM with binomial distribution and logit link function. Also, GLM with logit distribution would be a more robust test for percentages. Why was mortality/survival not analyzed using survival analysis?

Thanks for the suggestions, we have incorporated them where applicable. *C. carnea* choice between prey from different diets (Figure 6E) was analyzed with a two-sided binomial test, and effects between each treatment were now analyzed via GLM with a binomial distribution and a logit link function. The *P. xylostella* egg hatching percentages (Figure 2E, F, Figure 4E, F, and Figure 2—figure supplement 2E-H) were now also analyzed by GLM with binomial distribution and logit link function. The results of mortality (Figure 2B, Figure 4B, and Figure 6C) and predation assays (Figure 6F, G) were evaluated using two-sided proportions test to analyze the percentage data. *C. carnea* pupation (Figure 6B) was additionally reanalyzed by Kaplan-Meier survival analysis. Statistical test results, including data from experiment repetitions, are shown in the corresponding Source Data files.

In some experiments, it was unclear whether caterpillars were being fed individually or in groups. It is important to know how many plants and how many caterpillars were used because caterpillar feeding on the same plant are not independent and should not be treated as such in the analyses.

We have incorporated additional information on the larval feeding in the Materials and methods section (subsections “Collection of *P. xylostella* tissues” and“Frass spiking assay”).

Badenes-Perez, F. R., Reichelt, M., Gershenzon, J., and Heckel, D. G., 2013. Interaction of glucosinolate content of *Arabidopsis thaliana* mutant lines and feeding and oviposition by generalist and specialist lepidopterans. Phytochemistry. 86: 36-43. doi:https://doi.org/10.1016/j.phytochem.2012.11.006

Li, Q., Eigenbrode, S. D., Stringam, G. R., and Thiagarajah, M. R., 2000. Feeding and growth of *Plutella xylostella* and *Spodoptera eridania* on *Brassica juncea* with varying glucosinolate concentrations and myrosinase activities. J. Chem. Ecol. 26: 2401-2419. doi:https://doi.org/10.1023/a:1005535129399

Renwick, J. A. A., Haribal, M., Gouinguené, S., and Städler, E., 2006. Isothiocyanates stimulating oviposition by the diamondback moth, *Plutella xylostella*. J. Chem. Ecol. 32: 755-766. doi:https://doi.org/10.1007/s10886-006-9036-9

[Editors' note: further revisions were requested prior to acceptance, as described below.]

Essential revisions:All the reviewers were happy with the revision but in discussions we all agreed that we should go with a revision to provide a chance to clarify some aspects surrounding the multi-modality. We agree that the biomodality is balanced across the groups so not changing the mean relationships. However, nearly every reviewer and editor has been puzzled by this indicating that every reader will be equally puzzled. Any explicit thoughts that you can provide to explain this biomodality will go a long way to alleviating the reviewers puzzlement. Different mother ages? etc? Further, it would be advised to state that the residuals are normally distributed between the groups in these tests to alleviate statistical concerns.

As previously stated, unfortunately we cannot clearly explain these results based on our observations, in spite of this distribution being consistent among the different experiments where egg hatching was recorded. Both parents were always newly emerged, so there were no differences in moth ages. Our current hypothesis is that this was an effect from the ‘forced’ monogamy during pairings, which prevents mate choice by the females and can result in incompatible mating pairs and a larger proportion of unfertilized eggs. We have added a few sentences explaining this possibility into the Discussion (subsection “Plant-mediated RNAi efficiently silences *P. xylostella gss* with severe physiological and fitness consequences”, second paragraph).

We would also like to thank the suggestion to check the normality of the residuals in this experiment. As could perhaps be expected from the bimodal distributions, the residuals of several groups were not normally distributed. To correct this, we have changed the analyses of these data to non-parametric Kruskal-Wallis tests followed by Dunn’s post-hoc tests, instead of GLM. We also added this information in the “Statistical analyses” subsection of the Materials and methods.

Reviewer #1:I would like to thank the reviewers for shifting to two-way ANOVAs but it is not clear how these are being presented in the figures. A two-way ANOVA will provide a significance for the main effects and the interaction while in the figures it still appears to be pairwise t-test results that are being shown.

Thank you for pointing this out. We have now changed the notation of statistical differences in these figures, and expanded the test results in the figure legend and source data files.

I am still concerned by the bimodality in the egg laying/hatching experiment and the correlation analysis further raises this as a concern. In the correlation analysis, it appears that there are really two correlations in the data with one being a low hatching. Is there variation in the age of the mothers such that some are either much younger or older than others that could be linked to this?

As mentioned in the response above, there was not a wide range of parental ages in these experiments, with moths being all newly emerged. While we unfortunately cannot explain this effect based on our results, we have added a short discussion of our current hypothesis for the bimodality of these data (subsection “Plant-mediated RNAi efficiently silences *P. xylostella gss* with severe physiological and fitness consequences”, second paragraph).